# Multitemporal glacier inventory revealing four decades of glacier changes in the Ladakh region

Mohd Soheb[1], Alagappan Ramanathan[1], Anshuman Bhardwaj[2], Millie Coleman[2,3], Brice Rea[2],

Matteo Spagnolo[2], Shaktiman Singh[2], Lydia Sam[2]

[1] School of Environmental Sciences, Jawaharlal Nehru University, India.

[2]School of Geosciences, University of Aberdeen, United Kingdom.

[3]School of Natural and Built Environment, Queen's University Belfast, United Kingdom.

*Correspondence to*: Mohd Soheb (sohaib.achaa@gmail.com)

**Abstract.** Glacier inventories, and changes therein, play an important role in understanding glacier dynamics and water resources over larger regions. In this study, we present a Landsat-based multi-temporal inventory of glaciers in four Upper Indus sub-basins and three internal drainage basins in the Ladakh region for the years 1977, 1994, 2009 and 2019. The study records data on 2257 glaciers (>0.5 km$^2$) covering ~7923 ±212 km$^2$ equivalent to ~30% of the glaciers and ~89% of the glacierised area within the region. Results show that the highest concentration of glaciers is found in the higher elevation zones, between 5000 and 6000 m a.s.l, with most of them facing north. The area and length of nearly all the glaciers (~97%) have decreased over the past 42 years, by 6.9% and 12% respectively. However, heterogeneity in glacier change was observed across the basins. Glaciers in Shayok Basin experienced the least deglaciation (~3.9%), whereas Leh (~23%) and Tsokar Basin (~26%) experienced the greatest deglaciation over the 42 years (1977-2019). The major factors contributing to such differences were temperature, precipitation and size of the glaciers, with most changes occurring as a result of a drier winters and warmer summers. During the observation period, the region showed a statistically significant (at $p<0.01$) increasing trend in mean annual, JJAS and winter temperatures.

## 1. Introduction:

Himalaya is the largest storehouse of snow and ice outside the Polar Regions. This large reserve of water plays a crucial role in the hydro-economy of the region (Bolch, 2019; Frey et al., 2014; Maurer et al., 2019; Pritchard, 2019). Any change to the cryosphere would have a direct impact on the hydrology, further influencing the communities downstream whose livelihood and economy relies on and are supported by the major river systems e.g., the Brahmaputra, Ganges and Indus, among others. Melt water from this region feeds millions of people including megacities like Delhi, Dhaka, Karachi, Kolkata and Lahore (Azam et al., 2021; Immerzeel et al., 2010; Singh et al., 2016). Recent studies have reported that Himalayan glaciers are retreating at an alarming rate (Azam et al., 2021; Bolch, 2019; Kääb et al., 2015; Maurer et al., 2019; Pritchard, 2019; Shean et al., 2020, among others) with glaciers



of the Western Himalayas showing less shrinkage than the glaciers of the central and eastern parts (Azam et al., 2021;
Shukla et al., 2020; Singh et al., 2016). Glaciers in the nearby Karakoram region display long-term irregular behaviour
with frequent glacier advances/surges and minimal shrinkage, which is yet to be fully understood (Azam et al., 2021;
Bhambri et al., 2013; Bolch et al., 2012; Kulkarni, 2010; Liu et al., 2006; Minora et al., 2013; Negi et al., 2021).
Glaciers of the Karakoram region experienced an increase in area post-2000, due to surge-type glaciers. In just the
upper Shayok valley, as many as 18 glaciers, occupying more than one-third of the glacierised area, showed surge-
type behaviour (Bhambri et al., 2011, 2013; Negi et al., 2021). However, not all regions have been analysed at the
same level of temporal and spatial detail. In particular, our knowledge of glacier dynamics and their response to
climate change is still incomplete in the cold-arid, high-altitude Ladakh region (~105,476 km$^2$) comprising both, the
Himalayan and Karakoram ranges.
The advent of remote sensing technologies has permitted the mapping and measuring of various glacier attributes even
in the absence of sufficient in-situ observations (Bhardwaj et al., 2015). Glacierised area estimations have often relied
on global and regional glacier inventories such as the Randolph Glacier Inventory (RGI), Global Land Ice Monitoring
from Space (GLIMS), Geological Survey of India (GSI) inventory and Space Application Centre India (SAC)
inventory, among others. However, given the large scale of these inventories, automated techniques are often
employed to map and calculate glacier extent, with differing levels of success. Additionally, varying quality of satellite
imagery acquired from different time periods are sometimes necessitated in high mountain areas, such as Ladakh.
Together, these two factors can lead to over- or under-estimation of glacier areas leading to erroneous information on
temporal change. Moreover, there is no available multi-temporal glacier inventory for the region, which can inform
us on the changes in the natural frozen water reserves which have put the water security of this entire cold-arid region
under significant stress during recent years. The residents of Ladakh have witnessed a decrease in agricultural yields,
the main driver of economic development of the region, due to a decrease in water resources (Barrett & Bosak, 2018).
The water scarcity together with an increase in tourism footprint (four times more tourists (327,366) in 2018 than
2010, a number that is more than the entire population of Ladakh) has led to a shift in livelihood from agriculture to
other commercial activities (Müller et al., 2020), though even the latter relies heavily on water resources. In order to
cope with water scarcity, some people of Ladakh have developed new water management techniques, commonly
known as 'ice reservoirs' or 'ice stupas', to supplement agricultural activities (Nüsser, et al., 2019a,b).
This study presents a new multi-temporal glacier inventory for the Union Territory of Ladakh, India, covering 42
years of change between 1977 and 2019. The inventories are entirely based on Landsat imageries acquired mostly
during late-summer with additional quality control provided through high-resolution PlanetScope and Google Earth
imagery. We further assess the rate of change of glaciers in response to the regional climate trends and other factors,
and establish a comparison with other inventories. This new dataset and analyses of glacier distribution and spatio-
temporal change will improve understanding of glacier dynamics and the impact of ongoing climate change on water
supplies in the Ladakh region, where water in the arid season is mostly supplied by glaciers. The dataset produced in
this study can be viewed here: https://www.pangaea.de/tok/8b9a6e7275b32019eab155e11a461866706fabf3
**(temporary link for the reviewers')**

**2.   Study Area:**

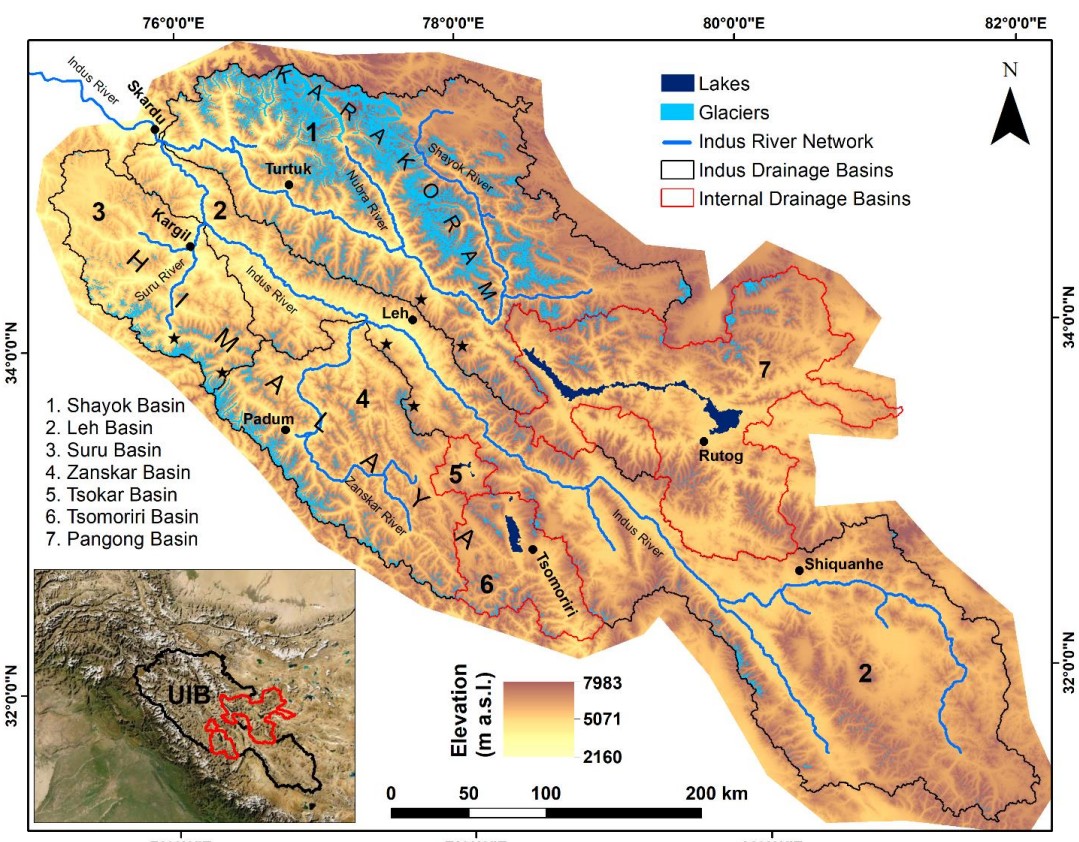


*Figure 1: Location map of the study area: the boundaries of studied Upper Indus Basin and internal drainage basins are outlined*
*in black and red on the digital elevation model (DEM) and in the inset map. Inset map shows the study area with respect to the*
*Himalayan and Karakoram region. Black dots and star represent the respective basins' major settlements and field investigated*
*glaciers. ASTER Global DEM was used to produce the base map.*

This study focuses on glaciers in the Upper Indus Basin (Figure 1) upstream of Skardu (hereafter UIB) and three
endorheic basins (or internal drainage basins, IDB hereafter) within the Ladakh region, namely Tsokar, Tsomoriri and
Pangong basins. The geographic extent of the study area lies within a latitude of 31.1º to 35.6º N and a longitude of
75.1º to 81.8º E and covers a vast region of the Karakoram and Western Himalayan ranges. UIB has an area of
~105,476 km$^2$, of which ~8302 km$^2$ (8%) is glacierised by ~6300 glaciers spanning elevations between ~3400 m and
~7500 m a.s.l. (as per RGI 6.0). IDBs of Tsokar (1036 km$^2$), Tsomoriri (5462 km$^2$) and Pangong (21,206 km$^2$) house
~30, 345 and 812 glaciers, comprising a glacierised area of ~7 (0.6%), 185 (3.4%) and 437 (2.1%) km$^2$, respectively
(as per RGI 6.0). The glaciers of IDBs are at a comparatively higher elevation, spanning from ~4800 to ~6800 m a.s.l.
Meltwater from these glaciers drains into the lakes within each basin. Pangong Lake (a saline lake), situated at an



elevation of ~4250 m a.s.l., is the largest with an area of ~703 km². Both Tsomoriri (freshwater lake at ~4530 m a.s.l.)
and Tsokar (saline lake at ~4540 m a.s.l.) Lakes are designated Ramsar sites which occupy areas of ~140 and ~15
km², respectively.
Ladakh has a cold-arid climate due to the rain shadow and elevation effects of the Himalaya and Karakoram mountains
(Schmidt & Nüsser, 2017). Mean annual air temperature and annual precipitation range between -20 to 10 ℃ and 0 to
1800 mm, respectively (Hersbach et al., 2020). This region is inhabited by ~700,000 people (as per Census of India
2011, Census of China 2019), most of which are directly, or indirectly, dependent on snow and glacier meltwater to
support hydropower generation, irrigation and domestic needs.

## 3.   Data and methods
### 3.1. Data

This study utilises multiple Landsat level-1 precision and terrain (L1TP) corrected scenes (63 scenes in total) from
four different periods: 1977±5 (hereafter 1977), 1994±1 (hereafter 1994), 2009 and 2019±1 (hereafter 2019). Scenes
from the 1970s are majorly (14 out of 17) from the year 1972 to 1977, however due to higher cloud cover and less
availability of imagery during the earlier Landsat period, three scenes from 1979 and 1980 were also included (Table
S1). Images from the late in the ablation season (July-October), having least snow and cloud cover (<30% overall,
and not over the glacierised parts), were selected and used for glacier identification and boundary delineation.
Advanced Space borne Thermal Emission and Reflection Radiometer Global Digital Elevation Model (ASTER DEM)
scenes were also used for basin delineation and calculating slope, aspect and elevation metrics of the glaciers. Glacier
digitisation, basin delineation and calculation of area were all performed in ArcGIS 10.4. Details of the imagery used
in this study are presented in (Table 1 and Table S1).
Additionally, long-term hourly ERA5 single level reanalysis temperature and precipitation data (Hersbach et al., 2020;
last accessed on 30 June 2020) have also been used to understand the regional climate and its evolution over the four-
decade window of interest. ERA5 is a fifth-generation ECMWF (European Centre for Medium-Range Weather
Forecasts) reanalysis dataset for global climate, at a high resolution (0.25°) and it has been shown to perform well for
the Ladakh region (Kanda et al., 2020). In-situ climate data from Leh (India, Indian Meteorological Department) and
Shiquanhe (China, https://www.ncei.noaa.gov/) were also used to bias correct the reanalysis dataset. Available time-
series datasets from the two stations (Leh and Shiquanhe) vary temporally. Leh station has around 27 years of data,
due to gaps ranging from months to years (Soheb et al., 2020) whereas, the Shiquanhe station comprises 38 years of
data for 1979 to 2019.





*Table 1: Information on the satellite imagery used in this study (Detailed info. in Table S1).*

| Dataset | Year of Acquisition | Spatial Resolution | Source | Purpose |
|---|---|---|---|---|
| Landsat MSS | 1977±5 | 60m | https://earthexplorer.usgs.gov/ | Glacier area mapping |
| Landsat TM | 1994±1, 2009 | 30m | | |
| Landsat OLI | 2019±1 | 15m | | |
| ASTER GDEM | 2000-2013 | 30m | https://earthdata.nasa.gov/ | Topography and basin delineation |

**3.2. Basin delineation**
Basin delineation was carried using ASTER GDEM V003 and the Hydrology tool in ArcGIS. The input DEM was
first analysed to fill-in all sinks with careful consideration of the potential for basin area over-estimation (Khan et al.,
2014). UIB was delineated using a pour point advertently selected at the Indus River in Skardu as we aimed to assess
all the tributary basins of the Ladakh region. The UIB obtained by this approach was further divided into second-order
tributary basins, i.e., Shayok, Suru, Zanskar and Leh basins. A small portion of the leftover area from UIB, after
second-order tributary basin delineation, was merged into the Leh basin so that the entire UIB, upstream of Skardu,
was investigated. Delineation of the three endoreic basins (IDBs) that lie partially or completely in the Ladakh region,
i.e., Tsokar, Tsomoriri and Pangong basins, was also carried out using the same method with the help of respective
lakes as a pour point.
**3.3. Glacier mapping**
Glaciers were mapped using a two-way approach, closely following the Global Land Ice Measurements from Space
(GLIMS) guidelines (Paul et al., 2009): 1) automatic mapping of the clean glacier and 2) manually correcting the
glacier outlines and digitisation of debris cover. First, a band ratio approach between NIR (Near Infrared) and SWIR
(Shortwave Infrared) (as suggested by Paul et al., 2002, 2015; Racoviteanu et al., 2009; Bhardwaj et al 2015; Schmidt
& Nüsser, 2017; Smith et al., 2015; Winsvold et al., 2014, 2016) with a threshold of 2.0 (NIR/SWIR > 2 = ice/snow)
was used on 2019 Landsat OLI images to delineate the clean part of glaciers. A median filter of kernel size 3 x 3 was
applied to remove the isolated and small pixels outside the glacier area. The NIR and SWIR band ratio approach is
good at distinguishing glacier pixels from water features with similar spectral reflectance values (Racoviteanu et al.,
2009; Zhang et al., 2019). This approach failed in areas with high snow/cloud cover, shadows, frozen channels/lakes
and debris cover. The snow/cloud cover and frozen lakes/stream problem were addressed by selecting Landsat scenes
from the ablation period (July-October) with the cloud cover < 30%. The issue with the snow-covered regions in
accumulation zones, where the delineation was the most challenging, was resolved using the best available imagery
of any time between 1977 and 2019 because glaciers are not expected to change their shape significantly in the higher
accumulation zones. One of the major issues was the debris covered glaciers, which had to be manually digitised, with
the support of high-resolution Google Earth and PlanetScope imagery from 2019 ±2. The result was then used as a
basis for manual digitisation of debris covered glaciers in other years where high-resolution images are not available.



In most cases, identification of the glacier terminus was made with certain contextual characteristics at the snout, e.g.,
the emergence of meltwater streams, proglacial lakes, ice walls, end moraines etc. (Figure S1).
The glacier outlines from 2019 were used as a starting point for the subsequent digitization of glacier areas of 2009,
1994 and 1977. Glacier length was measured using a semi-automatic approach, by employing the DEM to identify a
central flow line for each mapped glacier (Ji et al., 2017). Further manual corrections were undertaken to account for
the flow lines of glaciers that have multiple tributaries and multiple highest/lowest points. Furthermore, some mapping
errors are still expected to be present in this inventory due to a possible misinterpretation of glacier features, and the
quantification of such errors are difficult owing to the lack of reliable reference in-situ data in the Ladakh region. Such
errors were minimized by keeping a fixed map-scale of 1:10000 in most cases and doing a quality check on glacier
outlines using high-resolution images.
Other specific glacier attributes were also extracted including new glacier Ids, Global Land Ice Measurements from
Space (GLIMS)-Ids, Randolph Glacier Inventory (RGI 6.0)-Ids, coordinates (latitude and longitude), elevation
(maximum, mean and minimum), aspect (mean), slope (mean), area and length (maximum).

**3.4. Uncertainty**
The uncertainties associated with the glacier outline mapping originate from the spatial resolution of the satellite
images and the misleading effect of seasonal snow, shadow, debris and cloud cover. Due to the lack of ground truth
data for glaciers in the Ladakh region uncertainty estimation was performed following Paul et al., (2017). We applied
a buffer based assessment, with the buffer width set to one-pixel for debris and half-pixel for clean ice (Bolch et al.,
2010; Granshaw & G. Fountain, 2006; Mölg et al., 2018; Tielidze & Wheate, 2018), given that the level 1TP Landsat
images were corrected to sub-pixel geometric accuracy (Bhambri et al., 2013).
The associated uncertainty for smaller glaciers (<0.5 km$^2$) amounts to ~15-30%, so all glaciers with an area of less
than 0.5 km$^2$, which comprise ~70% and ~10% of the total glacier count and glacierised area, are not included in this
study. For the remaining glaciers, the uncertainty ranged between ±2.7 and ±10.3% depending on the spatial resolution
of the satellite imagery and the individual glacier size. The highest uncertainty was for the year 1977 due to the coarser
spatial resolution of Landsat MSS data when applied to the smallest glaciers (0.5-1 km$^2$). Overall, the uncertainty was
found to be 6% (Table S2).
**3.5. Climate data correction and analysis:**
Kanda et al., 2020 evaluated the performance of seven gridded climate datasets (e.g., APHRODITE, CRU-TS, PGF,
UDEL, ERAI etc.) against observed data from the 19 in-situ stations scattered across the Karakoram, Greater and
Lower western Himalaya. It was found that ERA-Interim, with additional bias corrections, performed best. Since the
ERA5 is the improved version of ERA-Interim (Hersbach et al., 2020), it was further bias corrected and used to
understand the regional trends in temperature and precipitation, the first order driver of surface mass balance. Negi et
al., 2021 used 12 in-situ meteorological stations from the Shayok region and found the mean summer, mean winter
and mean annual temperature to be 3, -14 and -6.9 ⁰C, respectively, between 1985 and 2015. The ERA5 reanalysis
data from the Shayok basin are in agreement with the in-situ temperatures.

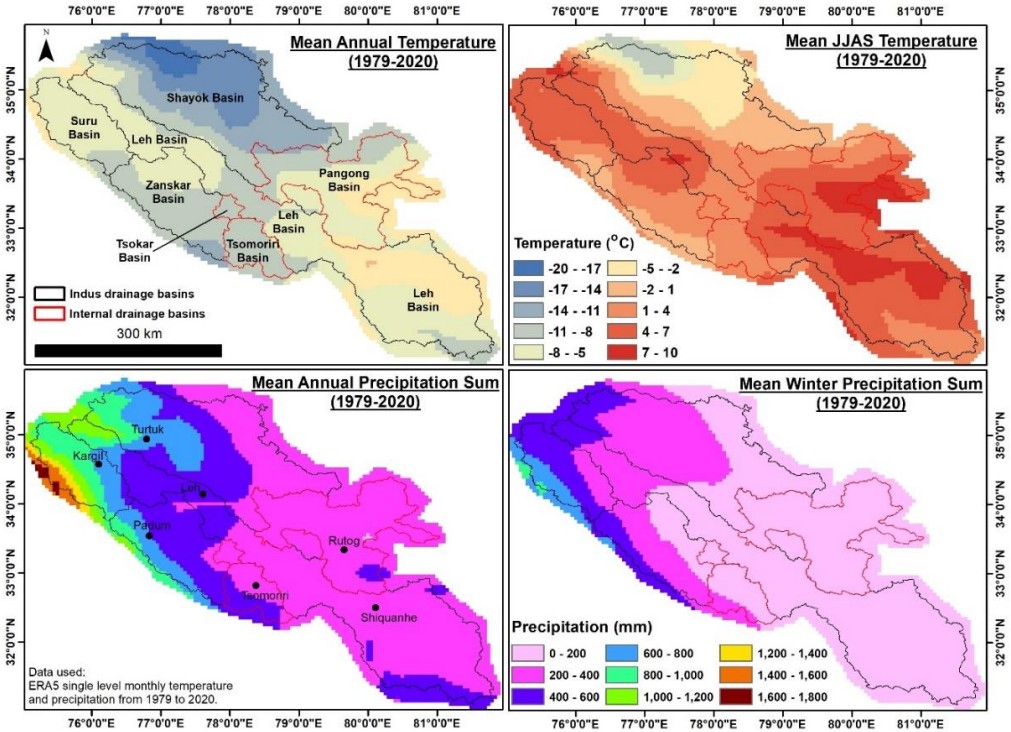


*Figure 2: Distributive temperature (annual and JJAS) and precipitation (annual and winter) across the study area. Data used:*
*ERA5 single level monthly temperature and precipitation from 1979 to 2020, https://climate.copernicus.eu/climate-reanalysis*
The linear scaling bias correction method (Ines & Hansen, 2006; Shrestha et al., 2017; Teutschbein & Seibert, 2013)
was adopted for the corrections with the help of the available observed long-term Leh and Shiquanhe datasets. Only
seven grids from the entire region were bias corrected because of the limited observed dataset. One grid cell containing
a major village/town/city from each basin was chosen except for Tsokar and Leh basins. For the Leh basin two grid
cells were chosen because of the available observed records from Leh and Shiquanhe stations, whereas no grid cell
was chosen from the Tsokar basin due to its comparatively smaller area (Figure 1 and 2). The bias corrected data from
seven grids were further analysed statistically to understand the significance and magnitude of annual and seasonal
trends in temperature, precipitation and positive degree days (PDDs) using the Mann-Kendall test and Sen's slope
estimator (Sen, 1968), respectively. The change in temperature, precipitation and PDD was calculated following
Shukla et al., 2020.





# 4. Results

## 4.1. Glacier inventory of 2019

*Table 2: Basin-wide glacier information of UIB and IDBs based on present study for the year 2019.*

| Basin | Basin area km² | Total Area > 0.5 km² Count | Total Area > 0.5 km² Area km² | Area 0.5-1 km² Count | Area 0.5-1 km² Area km² | Area 1-5 km² Count | Area 1-5 km² Area km² | Area 5-10 km² Count | Area 5-10 km² Area km² | Area 10-50 km² Count | Area 10-50 km² Area km² | Area 50-100 km² Count | Area 50-100 km² Area km² | Area > 100 km² Count | Area > 100 km² Area km² |
|---|---|---|---|---|---|---|---|---|---|---|---|---|---|---|---|
| All | 132180 | 2257 | 7923 | 980 | 694 | 1053 | 2206 | 124 | 853 | 84 | 1617 | 10 | 674 | 7 | 1879 |
| Shayok | 33579 | 1268 (56%) | 5864 (74%) | 495 (51%) | 351 (51%) | 609 (58%) | 1304 (59%) | 88 (71%) | 621 (73%) | 60 (71%) | 1151 (71%) | 8 (80%) | 559 (83%) | 7 (100%) | 1879 (100%) |
| Leh | 46579 | 247 (11%) | 334 (4%) | 147 (15%) | 105 (16%) | 95 (9%) | 191 (9%) | 4 (3%) | 26 (3%) | 1 (1%) | 12 (1%) | 0 | 0 | 0 | 0 |
| Suru | 10502 | 201 (9%) | 498 (6%) | 81 (8%) | 59 (9%) | 100 (9%) | 212 (10%) | 12 (10%) | 69 (8%) | 8 (10%) | 159 (10%) | 0 | 0 | 0 | 0 |
| Zanskar | 14817 | 256 (12%) | 775 (10%) | 116 (12%) | 82 (12%) | 111 (11%) | 235 (11%) | 15 (12%) | 108 (13%) | 12 (14%) | 235 (15%) | 2 (20%) | 115 (17%) | 0 | 0 |
| Tsokar | 1036 | 3 (0.1%) | 3.5 (0.04%) | 2 (0.2%) | 1.5 (0.2%) | 1 (0.1%) | 2 (0.1%) | 0 | 0 | 0 | 0 | 0 | 0 | 0 | 0 |
| Tsomoriri | 5462 | 94 (4%) | 135 (2%) | 47 (5%) | 22 (3%) | 46 (4%) | 95 (4%) | 1 (1%) | 7 (1%) | 0 | 0 | 0 | 0 | 0 | 0 |
| Pangong | 21206 | 190 (8%) | 315 (4%) | 92 (9%) | 63 (9%) | 91 (9%) | 168 (8%) | 4 (3%) | 22 (2%) | 3 (4%) | 60 (4%) | 0 | 0 | 0 | 0 |



208

DATA: 2019,  GLACIERS > 0.5 km²
a. Count: 2257,  Area: 7923 ±212 km²
b. Count: 1267,  Area: 5864 ±151 km²
c. Count: 246,  Area: 333 ±11 km²
d. Count: 201,  Area: 499 ±14 km²
e. Count: 256,  Area: 775 ±23 km²
f. Count: 3,  Area: 3.4 ±0.1 km²
g. Count: 94,  Area: 136 ±5 km²
h. Count: 190,  Area: 313 ±10 km²

209

Figure 3: General statistics of the glaciers in UIB and IDBs: orientation of glaciers (i) and associated area distribution (ii),
Maximum, minimum and mean elevation of glaciers (iii), hypsometry of glacierised area (iv) and slope against glacier area (v)
and elevation (vi).

In 2019 the number of glaciers (>0.5 km²) is 2257 with an area totaling 7923 ±212 km². The glacierised area amounts
to ~6% of the overall region with glacier areas and lengths ranging between 0.5 to 862 km² and 0.4 to 73 km,
respectively. Shayok Basin and glacier category of 1-5 km² occupies the highest glacierised area, whereas Tsokar
Basin and glacier category of >100 km² comprise the least glacierised areas. Around 74% (1665) of the glaciers face

the northern quadrant (NW-N-NE) amounting to ~50% (3940 km$^2$) of the glacierised area. However, the orientation
and respective area coverage of glaciers varies within individual basins (Figure 3). Small glaciers were mainly found
to occupy the higher elevations above 5500 and vice versa. The mean elevation of the glaciers for the entire study was
~5587 m a.s.l. and the majority (73%, 5810 km$^2$) of the glacierised area is located between 5000 and 6000 m a.s.l.
(Figure 3). The mean slope of these glaciers ranged between 8 and 46º, and was found to decrease with increasing
glacier area. Glaciers with area greater than 100 km$^2$ have comparatively the lowest mean slope of 13º whereas, higher
mean slopes (23º) were found for smaller glaciers. Overall, the mean glacier slope was ~21º (Figure 3).

**4.2. Total glacier change between 1977 and 2019**

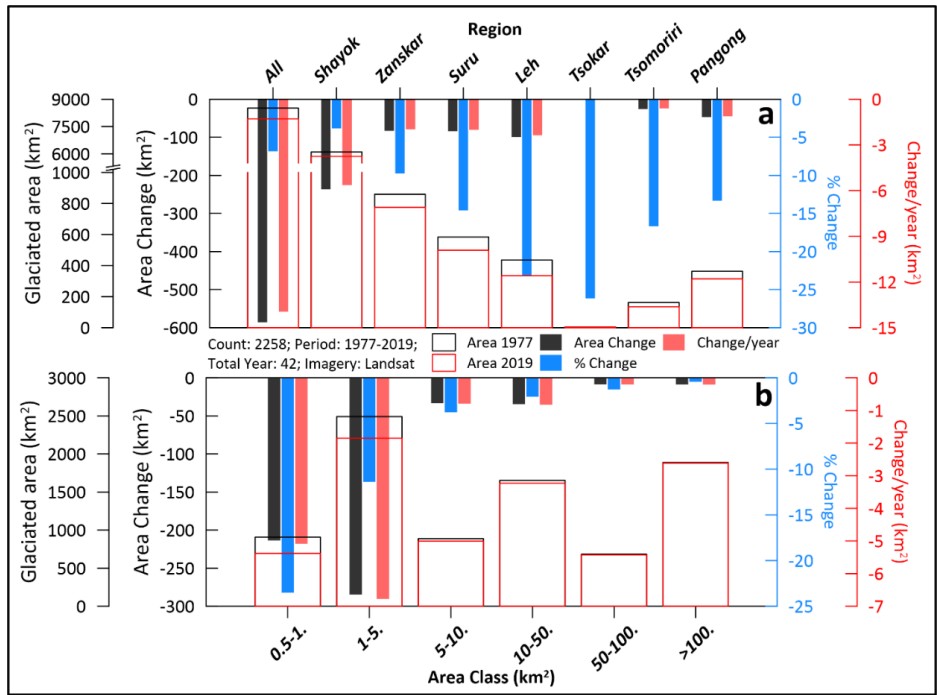

*Figure 4: Total change in the glacierised area with respect to a) basin and b) area class in 42 years (1977-2019).*
The total glacierised area reduced from 8511 ±861 km$^2$ in 1977 to 7923 ±212 km$^2$ in 2019, revealing an overall change
of ~-588 km$^2$ (-6.9%) in 42 years with a mean change of ~-14 km$^2$ year$^{-1}$ (-0.2% year$^{-1}$). The area change was found
to be different for the individual UIB sub-basins and IDBs. Relative change in these basins ranged between -3.9 and
-26%, with the least and greatest change found in Shayok and Tsokar basins, respectively (Figure 4). Glaciers in
Shayok basin witnessed a change of ~-238 km$^2$ (~-5.7 km$^2$ year$^{-1}$) from 6102 ±595 km$^2$ in 1977 to 5864 ±151 km$^2$ in
2019. Whereas, glacierised area of Tsokar basin reduced from 4.6 ±0.5 km$^2$ (1977) to 3.4 ±0.1 km$^2$ (2019), exhibiting
a change of ~-1.2 km$^2$ (~-0.03 km$^2$ year$^{-1}$). Glacierised area in other basins also witnessed change ranging between ~-
10 and -16% over 42 years.
Deglaciation associated with different area classes ranged between -0.5 (>100 km$^2$) to -24% (0.5-1 km$^2$). Our results
show that the highest relative change was mostly observed in the two smallest area classes of 0.5-1 and 1-5 km$^2$
whereas in larger area classes the change was found to be less than 4%. In the area category of 0.5-1 km$^2$, where the
maximum change happened, glacierised area reduced from 909 ±161 km$^2$ (1977) to 694 ±34 km$^2$ (2019), exhibiting a
change of ~-214 km$^2$ (-2.82 km$^2$ year$^{-1}$). The glaciers >100 km$^2$ witnessed a change of ~-10 km$^2$ (-0.22 km$^2$ year$^{-1}$)
from 1977 (1888 ±115 km$^2$) to 2019 (1879 ±30 km$^2$).

**4.3.** **Periodical change in glacierised area between 1977 and 2019**

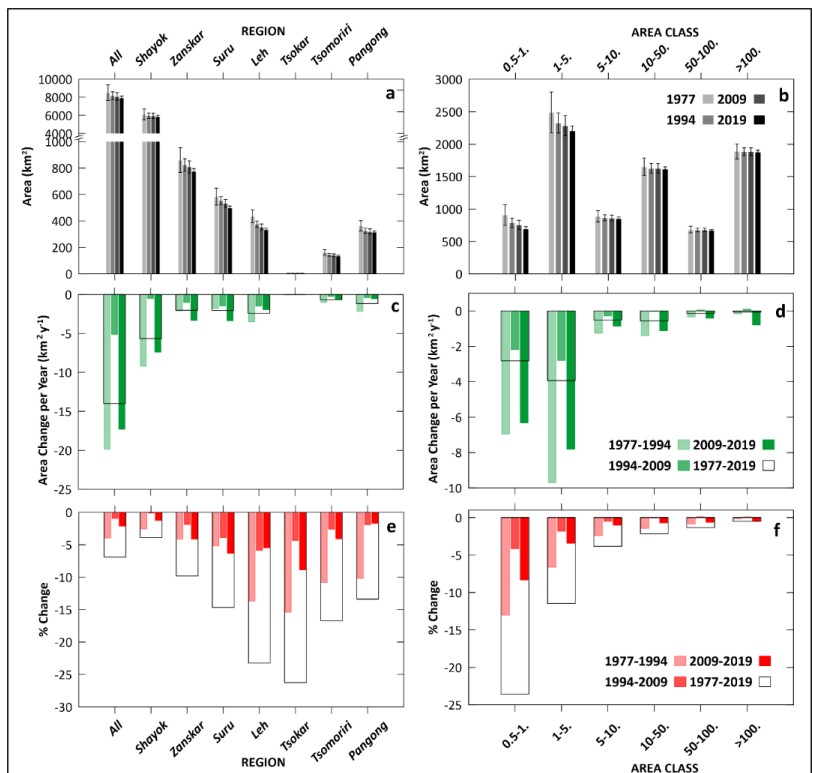

*Figure 5: Periodical change in glacierised area: basin-wise (a, c, e) and area classes (b, d, f), between1977 and 2019.*

The results show that the deglaciation rate was variable across the measurement windows (1977-1994, 1994-2009 and
2009-2019). The highest rate of change was found during the first period (1977-1994; -20 km$^2$ year$^{-1}$; -0.23% year$^{-1}$)
followed by the third (2009-2019; -17 km$^2$ year$^{-1}$; -0.21% year$^{-1}$) whereas the second period (1994-2009) witnessed
the lowest rate of change at ~-5 km$^2$ year$^{-1}$ (-0.06% year$^{-1}$). A similar pattern of changes was found in most of the UIB
sub-basins and in the three IDBs. But for the Zanskar and Suru basins, the rate of change was found to be highest in
the third period (2009-2019) with -3.34 (-0.41% year$^{-1}$) and -3.37 km$^2$ year$^{-1}$ (-0.63% year$^{-1}$), respectively (Figure 5).
The maximum rate of change was found in the glaciers of 1-5 km$^2$, ~-10, -3 and -8 km$^2$ year$^{-1}$, equivalent to -0.77, -
0.28 and -0.83% year$^{-1}$ during 1977-1994, 1994-2009 and 2009-2019, respectively. The lowest rate of change was
observed in the larger, >100 km$^2$, glaciers with a change of -0.2 (-0.01% year$^{-1}$), +0.1 (+0.01% year$^{-1}$) and -0.8 (-
0.04% year$^{-1}$) km$^2$ year$^{-1}$ during first, second and third periods. The period 1977-1994 witnessed the maximum
deglaciation per year in all area classes except the two largest classes where maximum deglaciation happened during
the third period (2009-2019). The lowest rate of change was observed during the second period (1994-2009) where
the two largest area classes (50-100 and >100 km$^2$) experienced a positive mean change of +0.05 and +0.1 km$^2$ year$^{-1}$
equivalent to ~+0.01 and +0.01% year$^{-1}$, respectively. Some examples from the Shayok Basin of different type of
glacier change i.e., surging, retreating, stable and change in high debris covered glacier are presented in Figure 6.

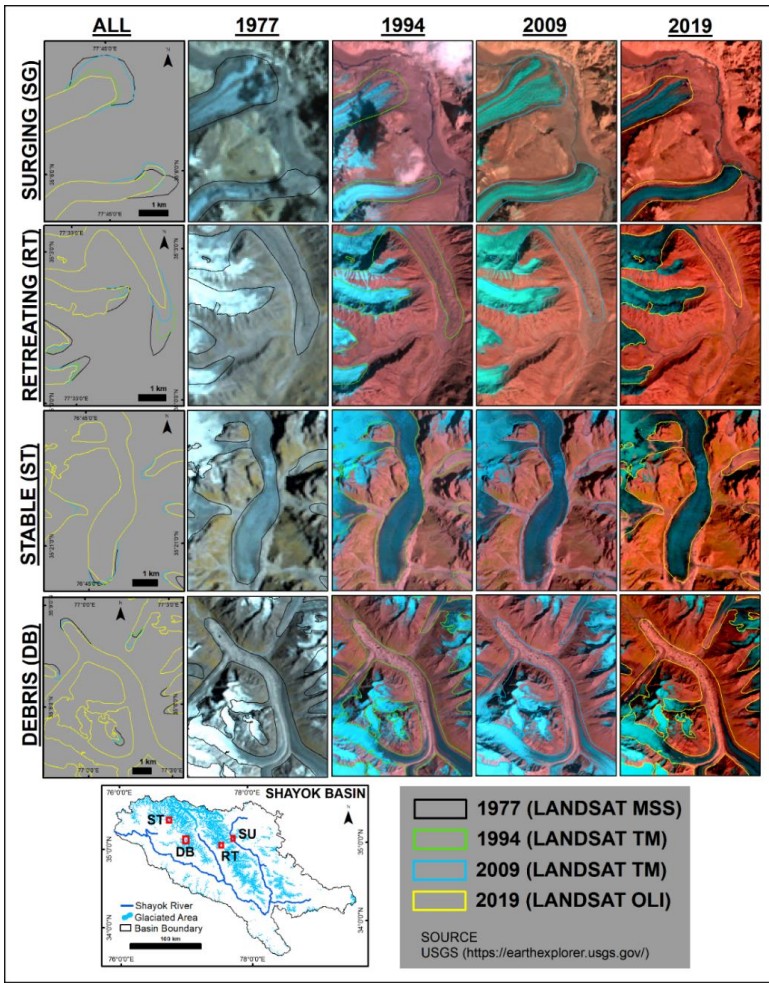


*Figure 6: Examples of different types of glacier change from the Shayok basin: surging type, retreating, stable and high debris*
*covered glacier between1977 and 2019.*

Earth System
Science
Data

**4.4. Glacier length change between 1977 and 2019.**

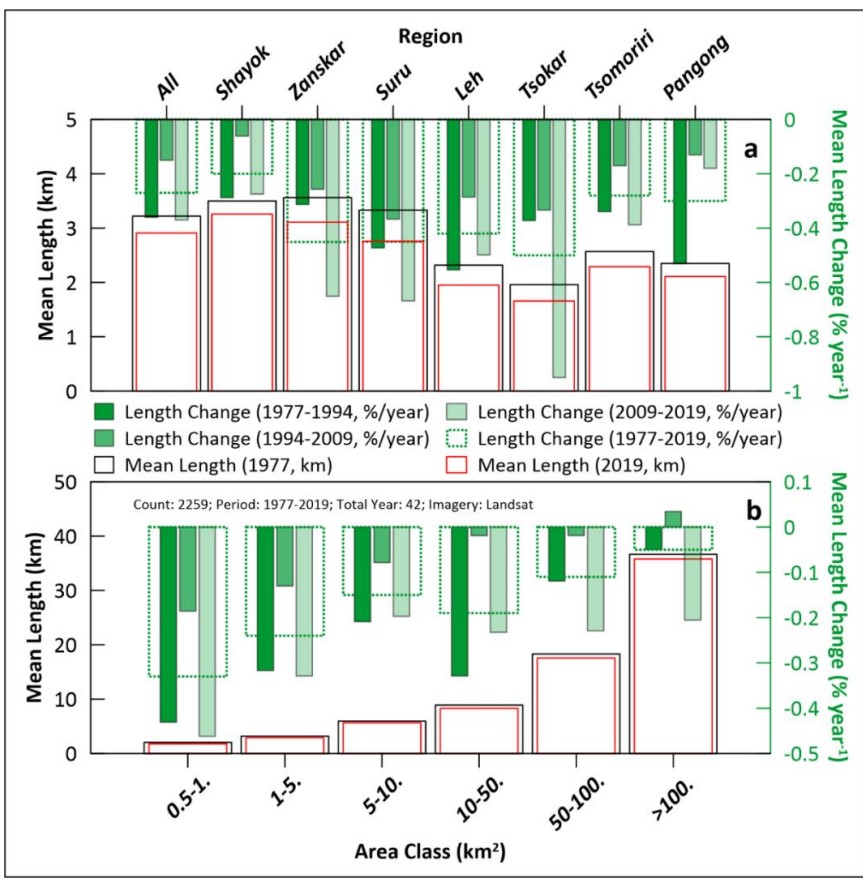


*Figure 7: Temporal rate of change of glacier length in study area, basins and area classes.*

Glacier lengths between 1977 and 2019 show a change ranging between +13 to -59% with the majority of retreat in
the range of ~-5 to -30%. It was found that nearly all (97%) glaciers retreated over the measurement window with a
mean retreat of ~-12% (-0.27% year⁻¹). Tsokar, Suru and Leh basin witnessed a comparatively higher rate of change
~-19, -19 and -18% equivalent to -0.5, -0.5 and -0.4 % year⁻¹, respectively. Whereas, change was relatively lower in
Shayok (-8%), Zanskar (-15%), Tsomoriri (-12%), Pangong (-12%) and Shayok (-8%) from 1977 to 2019. For the
area categories, the total length change ranges between -2.5 to -14% with the highest change found for the smallest
area class (0.5-1 km²) and vice versa.
The mean rate of change in glacier length was temporally variable, -0.4, -0.2 and -0.4% year⁻¹ for 1977-1994, 1994-
2009 and 2009-2019, respectively. The retreat rate for 1977-1994 was highest in Shayok, Leh and Pangong basins,
and in the two area classes (5-10 and 10-50 km²). Other basins and area classes witnessed the highest rate in length
change during the period 2009-2019. The rate was found to be lowest during 1994-2009 in all basins and area classes
(Figure 7) and a positive change was also observed in the largest area class, >100 km², during 1994-2009 as a result
of glacier surging.

**4.5. Meteorological conditions and their trends**

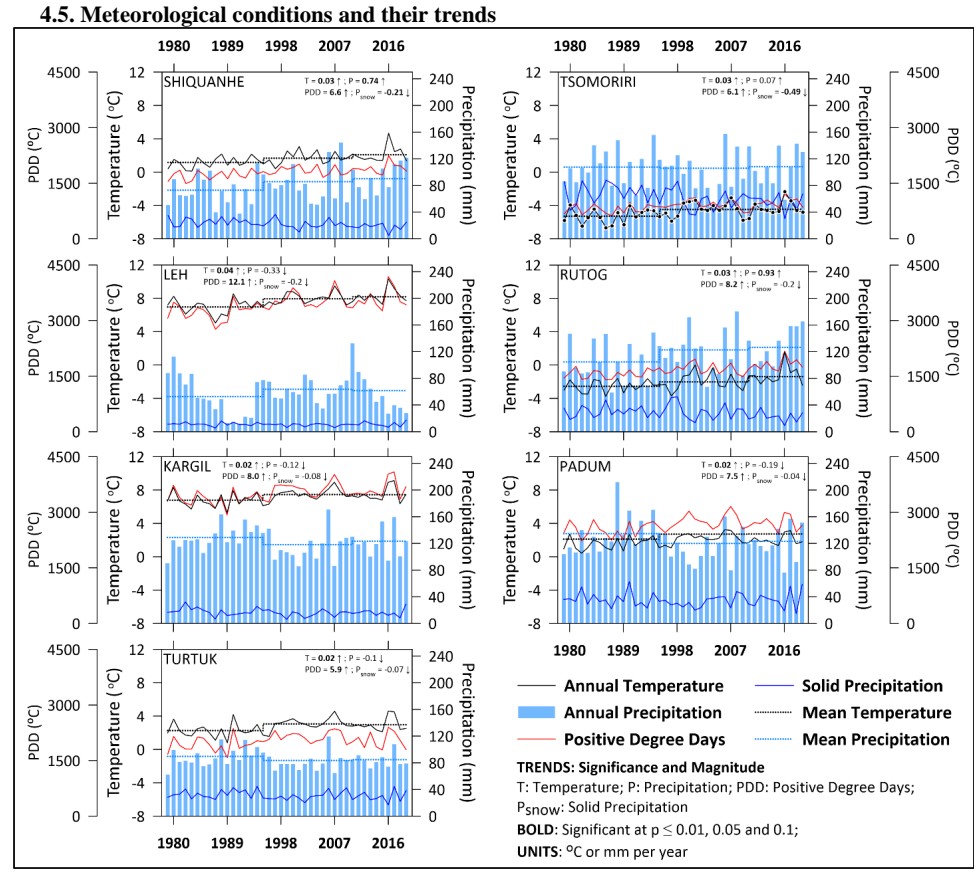


*Figure 8: ERA5 bias-corrected annual temperature (mean) and precipitation (sum) of the seven grids of the study area. Mean*
*temperature (dotted line) represent mean of the period and PDD is the sum of degrees above 0 ºC per day of the year.*

Meteorological conditions at the seven major settlements (Shiquanhe, Rutog, Leh, Turtuk, Tsomoriri, Padum and
Kargil) are presented in Figure 8 and Figure S3. Figure S2 presents the correlation between observed and bias-
corrected ERA5 reanalysis data. The result shows a strong correlation for temperature with R² = 0.97 (p<0.01) at both
Leh and Shiquanhe. A weaker but significant correlation was found for the precipitation at Shiquanhe (R² = 0.54, p<
0.01) and Leh (R² = 0.1, p < 0.01). The mean annual temperature and total annual precipitation in the study area,
during the 40-year period (1979-2019), ranged between -7 (Tsomoriri, 1986) to +10 °C (Leh, 2016) and 13 (Leh,
1991) to 213 mm (Padum, 1988), respectively.  The results show Leh and Kargil were comparatively the warmest,
with annual PDD sums exceeding 3000 °C while Tsomoriri was the coldest where the annual PDD sum never exceeds



1500 °C. Kargil, Padum, Tsomoriri and Rutog were wetter than other major settlements with Leh being driest (Figure
2 and 8). Solid precipitation, which mostly falls during winter months, was <80 mm in all the locations with highest
found in Tsomoriri (40 to 80 mm). Leh, Kargil and Shiquanhe had the lowest (< 40 mm) annual solid precipitation
(Figure 8).
Between 1979 and 2019 all locations showed a statistically significant increasing trend in mean annual, JJAS and
winter air temperatures with Leh exhibiting the highest rate of increase in mean annual, JJAS and winter temperatures
(Figure 8, Table S3). Padum had the lowest increase in temperature. Statistically significant trends in precipitation
were observed only in Shiquanhe (annual, JJAS), Rutog (annual, JJAS) and Tsomoriri (winter). Shiquanhe and Rutog
had positive trend in annual and JJAS precipitation and Tsomoriri had a negative trend in winter precipitation and
Tsomoriri and Shiquanhe experienced a decrease in solid precipitation (Figure 8, Table S3). A statistically significant
increasing trend in annual PDD sums was observed in all locations ranging between 5.9 and 12.1 °C year$^{-1}$, (Figure 8,
Table S3).

**5. Discussion**

**5.1. Description of the produced dataset and limitations**

The entire dataset of the multitemporal inventories of glaciers (>0.5 km$^2$) in Ladakh region for the year 1977, 1994,
2009 and 2019 is available at PANGAEA portal (https://doi.org/10.1594/PANGAEA.940994; Soheb et al., 2022).
The dataset are provided in two different GIS-ready file formats, i.e., GeoPackages (*.gpkg) and Shapefiles (*.dbf,
*.prj, *.sbn, *.sbx, *.shp, *.shx) to support a wider end users. GeoPackage is a relatively new and open file format
which is now being widely used and supported, whereas Shapefile format is one of the most widely used proprietary
but open file formats for vector datasets, supported by open-source GIS tools such as QGIS. The provided outlines of
glaciers, basins and lakes are all referenced to the WGS 84 / UTM zone 43N datum. For each region, there is one file
for basin and lake (if any) outlines, and four files for glacier outlines of 1977, 1994, 2009 and 2019. Each glacier
outline file contains glacier Ids (Jawaharlal Nehru University and University of Aberdeen glacier Ids, Randolph
Glacier Inventory 6.0 Ids, and Global Land Ice Measurements from Space initiative Ids), coordinates (latitude and
longitude), elevation (maximum, mean and minimum), aspect (mean), slope (mean), area and length (maximum).
While using this dataset, it is important to understand the key limitations of such regional-scale glacier inventories.
Some of the key user limitations of the produced dataset are: (1) Glaciers smaller than 0.5 km$^2$ (which comprise ~70%
and ~10% of the total glacier population and glacierised area, respectively) were not included in this inventory due to
the higher uncertainty (~15-30%) associated with these glacier outlines; (2) Inventories produced in this study are
entirely based on the medium resolution Landsat imageries, in the same way as other global or regional-scale glacier
inventories. Although the uncertainty associated with these inventories do not considerably impact regional-scale
analyses, care should be taken while using this data for a small set of glaciers. It should also be noted that it is not
feasible to produce multitemporal inventories regionally using high-resolution datasets due to scarce availability and
high costs of such high-resolution datasets; (3) The inventories of 1977±5, 1994±1 and 2019±1 are produced using
images with a range of acquisition dates due to the lack of data continuity within a particular year (more details in





section 3.1); and (4) The time periods chosen in this study are based on the availability of datasets and sufficient
temporal gaps among the datasets to allow multi-temporal glacio-hydrological analyses for a user.

### 337     5.2. Significance of the present inventory

The glacier inventory presented here has several improvements compared to the existing regional and global
inventories. Firstly, it covers the glaciers (> 0.5 km$^2$; n = 2257; ~7923 ±212 km$^2$) for the entire Ladakh region with
manual correction and quality control undertaken using freely available high-resolution images. The analyses were
further extended to estimate the change and distribution of ice masses at sub-basin scale. Secondly, for all the glaciers
a temporal rate of change has been calculated, which will aid hydrological and glaciological modelling aimed at
understanding past and future evolution. Finally, the new inventory will aid both the scientific community studying
the glaciers and water resources of the Ladakh region, and the administration of the Union Territory of Ladakh,
Government of India in developing efficient mitigation and adaptation strategies by improving the projections of
change on timescales relevant to policy makers.

### 347     5.2. Glacier and climate of the region

Overall, the majority (97%) of the glaciers in the Ladakh region retreated over the study period. The results,
unsurprisingly, also show a considerable heterogeneity. The Shayok Basin glaciers experienced the least change while
the Leh Basin deglacierised most. Some of the major factors contributing to such contrasting changes within this
region include climate (temperature and precipitation), topography, elevation, orientation, and glacier size. Most
documented changes appear to be occurring as a result of increasingly drier and warmer summers (lower precipitation
due to a weakened Indian Summer Monsoon) and wetter winters (particularly snow due to Western Disturbances)
(Farinotti et al., 2020; Yao et al., 2012). The moisture-laden ISM and WD (during summer and winter, respectively)
deliver precipitation mostly on the major orographic barriers of the Greater Himalayan and the Karakoram ranges, but
only limited precipitation bearing air masses reach the Ladakh region, specifically Leh and Tsokar basins resulting in
a greater deglaciation. However, glaciers of Suru, Zanskar, Tsomoriri and Pangong basins, which are proximal to the
major orographic barriers (Greater Himalayan and Karakoram ranges), showed comparatively moderate retreat.
The predominance of small glaciers in the Leh (97%), Tsokar (100%) and Tsomoriri (98%) basins is also a reason for
the contrasting behavior in glacier change. The smaller glaciers displayed the greatest deglaciation as they are more
sensitive to changes in surface mass balance (Yang et al., 2020). A small increase in ELA or SLA can turn these small
glaciers entirely into ablation zone, thus negatively impacting their health. Results also show that smaller glaciers
generally have steeper slopes thus experiencing a larger relative area loss than less steep glaciers.

### 365     5.3. Comparison of inventories in the Ladakh region

*Table 3: Basin and class wise comparison of the glacierised area between the present study and other inventories (RGI 6.0,*
*ICIMOD and GAMDAM).*

| Region | Present Study | RGI 6.0 | GAMDAM | ICIMOD |
| --- | --- | --- | --- | --- |

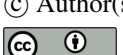



| | Area | Area | Difference | | Area | Difference | | Area | Difference | |
|---|---|---|---|---|---|---|---|---|---|---|
| | km² | km² | km² | % | km² | km² | % | km² | km² | % |
| **Shayok** | 5938 | 6999 | 1061 | 15 | 6616 | 678 | 10 | 5456 | -482 | -9 |
| **Zanskar** | 808 | 880 | 72 | 8 | 932 | 124 | 13 | 819 | 11 | 1 |
| **Suru** | 532 | 525 | -7 | -1 | 564 | 32 | 6 | 506 | -26 | -5 |
| **Leh** | 354 | 342 | -12 | -3 | 356 | 2 | 1 | 322 | -32 | -10 |
| **Tsokar** | 4 | 4.4 | 1 | 15 | 4.3 | 1 | 13 | 4.1 | 0 | 9 |
| **Tsomoriri** | 141 | 142 | 1 | 1 | 143 | 2 | 2 | 116 | -25 | -21 |
| **Pangong** | 320 | 320 | 0 | 0 | 335 | 15 | 4 | - | - | - |
| **Area Class** | | | | | | | | | | |
| **0.5-1** | 758 | 774 | 16 | 2 | 803 | 45 | 6 | 662 | -96 | -7 |
| **1-5.** | 2284 | 2437 | 153 | 6 | 2385 | 101 | 4 | 1958 | -326 | -12 |
| **5-10.** | 862 | 961 | 99 | 10 | 925 | 63 | 7 | 766 | -96 | -10 |
| **10-50.** | 1628 | 1959 | 331 | 17 | 1824 | 196 | 11 | 1356 | -272 | -20 |
| **50-100** | 678 | 730 | 52 | 7 | 599 | -79 | -13 | 592 | -86 | -15 |
| **>100** | 1887 | 2351 | 464 | 20 | 2412 | 525 | 22 | 1887 | 0 | 0 |
| **Total** | **8096** | **9212** | **1116** | **14** | **8950** | **854** | **11** | **7223** | **-533** | **-7** |


Differences in estimates of the glacierised areas are meaningful as they can lead to an over or under estimation of the available water resources. Therefore, correctly estimating glacier change over time is necessary for understanding glacier dynamics, future response to climate forcing and the water resources they provide. Table 3 presents a comparison between the present inventory and the Randolph Glacier Inventory (RGI) 6.0 (Pfeffer et al., 2014), International Centre for Integrated Mountain Development (ICIMOD) inventory (Bajracharya et al., 2019; Williams, 2013) and Glacier Area Mapping for Discharge in Asian Mountains (GAMDAM) inventory (Guo et al., 2015; Sakai, 2019), for the Ladakh region. The comparison involves glacier outlines for 2009 from the present study and excludes glaciers smaller than 0.5 km² from regional inventories to achieve the closest match temporally and for glacier size categories. Figure 9 presents a comparison of the only three inventories (present, RGI 6.0 and ICIMOD) on the five field-investigated glaciers of Ladakh region because RGI and GAMDAM inventory share the same outlines on these glaciers.

The comparison showed a higher glacierised area in RGI and GAMDAM inventories and lower in the ICIMOD inventory (Table 3) than the present inventory, with most of the differences contributed by the basins having the higher glacierised areas (Shayok and Zanskar) and from the larger glaciers (>10 km²). Such inconsistencies among the inventories are a product of several factors, e.g. 1) absence of change in glaciers over time due to the use of imagery with a wide range of acquisition years (Figure 9 a, c, d); 2) misinterpretation of the glacier terminus due to icing, debris, snow and cloud cover (Nagai et al., 2016), and 3) the methodology used. The smaller difference between the present and the ICIMOD inventory is mainly due to the adoption of a similar technique (i.e., a semi-automated approach) and the shorter time frame of the analysis that generated the ICIMOD inventory (i.e., 2002-2009).



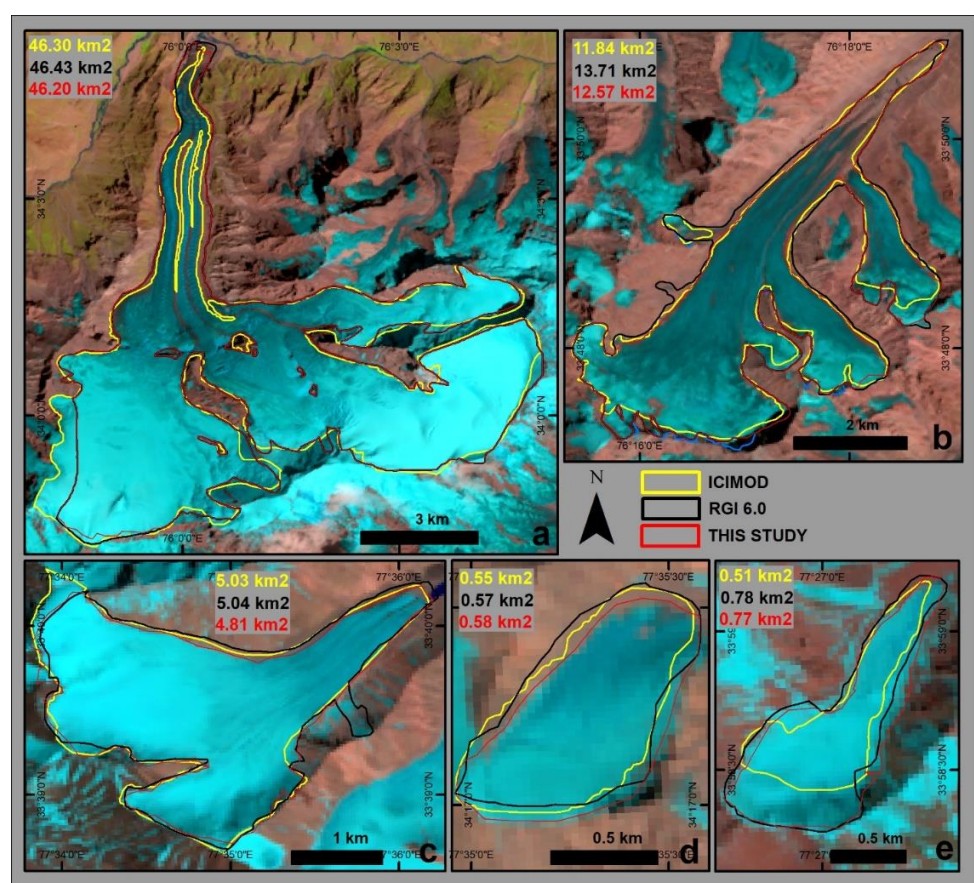


*Figure 9: Comparison of inventories on the field investigated glaciers of the study area: a) Parkachik glacier, Suru Basin; b)*
*Pensila glacier, Suru Basin; c) Lato glacier, Leh Basin; d) Khardung glacier, Shayok Basin; e) Stok glacier, Leh Basin.*

**5.4. Comparison with recent studies**
*Table 4: Comparison of glacier change attributes between the present study and others recent studies.*

| Region | Study | Period | Total Glaciers | Glacier Size (km²) | Area Change (km²) | Area Change (%) | Area Change /year (%) |
|---|---|---|---|---|---|---|---|
| Shayok Basin | Present | 1994-2009 | 1267 | >0.5 | -7.56 | -0.12 | -0.008 |
| | | 1994-2019 | 1267 | >0.5 | -81.69 | -1.4 | -0.05 |
| | Negi et al., 2021* | 1991-2014 | 569 | >1 | -7.8 | -0.91 | -0.008 |
| Leh Basin | Present | 1977-2019 | 247 | >0.5 | -102 | -23 | -0.5 |
| | Schmidt and Nüsser, 2017* | 1969-2016 | 135 | >0.03 | -21.79 | -19 | -0.4 |
| Suru Basin | Present | 1977-2019 | 201 | >0.5 | -85.78 | -14 | -0.3 |
| | Shukla et al., 2020* | 1971-2017 | 240 | >0.01 | -32 | -6 | -0.2 |

*\*Negi et al., 2021 basin area is comparatively small and within Indian territory only; \*Schmidt and Nüsser, 2017 studied temporal changes in*
*few selected glaciers; \*Shukla et al., 2020 studied the Suru sub-basin only.*





For the glaciers of the Shayok Basin our results agree with Negi et al., 2021, showing similar retreat rates (Table 4)
and lower summer, winter and annual temperatures reported over similar periods 1979-2019 (this study) versus 1985-
2015 (Negi et al., 2021). The lower summer temperatures and higher precipitation provide one explanation for the
area having the least change, apart from the 'Karakoram Anomaly', which is still not completely understood (Azam et
al., 2018, 2021; Bhambri et al., 2013; Hewitt, 2005; Minora et al., 2016; Negi et al., 2021). Another contributing factor
is likely due to higher proportion of surge-type glaciers which occupy a larger proportion of the glacierised area in the
Karakoram (Bhambri et al., 2013, 2017). Greater retreat in the glaciers in the Leh Basin (higher than the western
Himalayan average retreat rate of ~4 % year$^{-1}$, Shukla et al., 2020) appears to be driven by comparatively higher JJAS
temperatures and lower winter precipitation (Figure 2, 8). Leh Basin is narrow and long (~700 km) with relatively
high mean air temperature (both mean annual and JJAS) and low precipitation (total annual and winter precipitation)
in the region, especially in the area (Figure 2) where majority (~80%) of the glaciers are located. Retreat rates for
glaciers in the Leh basin from Chudley et al., 2017; Schmidt & Nüsser, 2012, 2017 and this study are all in agreement
(Table 4).
The relatively moderate retreat in the Suru and Zanskar Basins (Figure 4, 5, 7) is the result of a higher population of
large glaciers and meteorological conditions, particularly higher JJAS temperature and higher winter precipitation
(Figure 8) which appear to have somewhat cancelled each other out. Zanskar and Suru Basin results are also in line
with the recent finding by Maurer et al., 2019 and Shukla et al., 2020 which observed that the average mass loss had
doubled post-2000 i.e. this study: 1977-1994: -0.24, -0.31 % year$^{-1}$ and 2009-2019: -0.41, -0.63 % year$^{-1}$.
Unsurprisingly, the smaller glaciers are of greatest concern in relation to the implications for water resource provision,
given that they have shrunk the most over the past 42 years. Therefore, stress on water resources in the future is
expected to be greater, particularly around Leh Basin, where the majority of water resources derive from snow and
small glaciers melt only. The Leh Basin, situated between the Himalaya and Karakoram Ranges, displays a behaviour
more in line with the glaciers of Western Himalaya (Azam et al., 2018, 2021) and is significantly different from the
adjacent basins to the north (Shayok and Pangong) and to the south (Suru, Zanskar and Tsomoriri). Such a difference
is surprising yet understandable given the aridity and warmth of the climate in the valley (Figure 2, 8). This study
supports the findings by Chudley et al., 2017 and Schmidt & Nüsser, 2017 that the Leh Basin marks the transition
zone between the anomalous Karakoram, with its stable/surging glaciers, and the retreating Himalayan Range glaciers.

**6. Data availability**
For review purposes the data can be accessed via:
https://www.pangaea.de/tok/8b9a6e7275b32019eab155e11a461866706fabf3 (Soheb et al., 2022).
The dataset will be available for public after the completion of the review process.
The entire dataset of the Landsat based multitemporal inventory of glaciers, larger than 0.5 km$^2$, in Ladakh
region for the year 1977, 1994, 2009 and 2019 will be available at:
*PANGAEA*, https://doi.org/10.1594/PANGAEA.940994 (Soheb et al., 2022).





### 7. Conclusions

We compiled new glacier inventories of the UIB and IDBs within the Ladakh region for 1977, 1994, 2009 and 2019 using Landsat images. The inventory includes 2257 glaciers, larger than 0.5 km$^2$, covering an area of ~7923 ±212 km$^2$, which is ~14% and ~11% less than the RGI 6.0 and the GAMDAM, and 7% more than the ICIMOD inventories. The glaciers range in area between 0.5 to 862 km$^2$, with most of them belonging to the smallest size category (0.5-1 km$^2$) which account for ~694 km$^2$. The seven largest glaciers >100 km$^2$ account for the second largest glacierised area of ~1879 km$^2$. Shayok Basin hosts the highest number of glaciers and glacierised area; whereas, Tsokar Basin has the least. More than 70% of the glaciers are north-facing (NW-N-NE) and concentrated in higher elevation zones between 5000 and 6000 m a.s.l.

The study found that nearly all the glaciers (~97%) retreated and cumulatively lost a significant area of ~-588 km$^2$ (-6.9%) between 1977 and 2019. The retreat rates vary across the basins and glacier area categories. The relative area loss was highest in Leh, Tsokar and Tsomoriri Basins and for the small glacier categories. In the Shayok Basin and for the largest glaciers the relative area loss was lower. Smaller glaciers (0.5-1 and 1-5 km$^2$) have retreated most with a change of ~24 and ~11% (1977-2019). In other area categories, the retreat was less than 4%, with minimum change (~0.5%) found in the largest glacier area class of > 100 km$^2$. All basins showed a reduction in glacierised area of >10% except Shayok Basin, where the change was just 3.9% due to more favourable climatic conditions and the presence of surge-type and stable glaciers. The length change of individual glaciers was between +13 and -59% with a mean change of -12% (-0.27% year$^{-1}$) over 42 years.

Meteorological records show a statistically significant increasing trend in mean annual temperature, with the highest rate of +0.04 ºC year$^{-1}$observed in Leh. The annual precipitation trend was statistically significant only for Shiquanhe (+0.74 mm year$^{-1}$) and Rutog (+0.93 mm year$^{-1}$). An increasing trend in JJAS and winter temperature was observed for all locations, while an increasing trend in JJAS precipitation was found in Shiquanhe and Rutog only. Leh was the warmest and driest basin, with annual PDDs and precipitation of >3000 ºC and <120 mm, respectively. A statistically significant positive trend in PDDs was observed for all the locations, and a decreasing trend in solid precipitation was found in Shiquanhe and Tsomoriri.

The new multi-temporal inventory presented here will assist in planning the management of water resources, and for guiding scientific research focusing on glacier mass balance, hydrology and glacier change within the region. The detailed information and multi-temporal nature of this inventory will also aid in improving the existing global and regional inventories especially in the cold-arid Ladakh region where the majority of the population is highly dependent on glacier-derived melt water resources for domestic, irrigation and hydropower generation needs.

### Author contribution

MSo, AR, AB conceptualized and designed the study. MSo, AB and MC did the analysis. MSo wrote and AR, AB, MSp, BR, SS and LS edited the manuscript. All the authors have equally contributed to interpretation of the results.



**Acknowledgements**
The authors are thankful to the School of Environmental Sciences, Jawaharlal Nehru University, New Delhi, India,
for the lab facilities and the United States Geological Survey for the Landsat and ASTER imageries. The authors also
thank Planet Labs and Google for the high resolution PlanetScope and Google Earth imageries. We are also thankful
to the Scottish Funding Council and the University Of Aberdeen, United Kingdom for financially supporting our work.

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
