# Peer review of "Multitemporal glacier inventory revealing four decades of"

_Earth System Science Data, 2022_

## Author Comment (AC1)

**RESPONSE TO REFEREE#1**

We thank the reviewer#1 for the detailed review. We have carefully modified the manuscript according to the comments and suggestions. Below, we provide our responses (normal texts) to the comments (**Bold texts**) made by the reviewer#1. The *italics texts* are used to highlight the specific changes that were made in the manuscript.

The revised dataset can be viewed and downloaded using the following link: https://www.pangaea.de/tok/6d533482a662ef2124ed91eabdeec7b358dd8058

The study presents a multitemporal glacier inventory of Ladakh based on Landsat data for the years 1977, 1994, 2009 and 2019. The manuscript follows a clear structure, and most sections are well written. The study deals with large parts of the Upper Indus Basin (UIB) and three internal drainage basins (Tsomoriri, Tsokar and Pangong Tso) located in eastern Ladakh. The article is appropriate to support the publication of a useful and plausible data set. The data set solely contains Landsat data that can be used in future interpretations. However, I must mention several queries and specific comments.

**Abstract**

**1. The sentence "Glacier inventories, and changes therein, play an important role in understanding glacier dynamics and water resources over larger regions" (line 11-12) should be modified. The expression "changes therein" is not precise and it is obvious that glacier changes are important to understand glacier dynamics.**

Response: Thank you for the comment, we have now revised the sentence as per your suggestion, we have also revised the abstract as per the editors comment (Editors comment 1).

*"Multi-temporal inventories of glacierised regions provides an improved understanding of water resources availability. In this study, we present a Landsat-based multi-temporal inventory of glaciers in four Upper Indus sub-basins and three internal drainage basins in the Ladakh region for the years 1977, 1994, 2009 and 2019. The study records data on 2257 glaciers (of individual*

*size >0.5 km2) covering an area of ~7923 ±106 km2 which is equivalent to ~30% of the total glacier population and ~89% of the total glacierised area of the region. Glacier area ranged between 0.5±0.02 and 862±16 km2, while glacier length ranged between 0.4±0.02 and 73±0.54 km. Shayok Basin has the largest glacierised area and glacier population, while Tsokar has the least. Results show that the highest concentration of glaciers is found in the higher elevation zones, between 5000 and 6000 m a.s.l, with most of the glaciers facing towards the NW-NE quadrant. The error assessment shows that the uncertainty, based on buffer-based approach, range between 2.6 and 5.1% for glacier area, and 1.5 and 2.6% for glacier length with a mean uncertainty of 3.2 and 1.8%, respectively. This multitemporal inventory is in a good agreement with previous studies undertaken in parts of the Ladakh region. The new glacier database for the Ladakh region will be valuable for policy making bodies, and future glaciological and hydrological studies. The data can be viewed and downloaded from PANGAEA, https://doi.org/10.1594/PANGAEA.940994."*

**2: The study deals with 2257 glaciers larger than 0.5 km$^2$ covering 7923 ± 212 km$^2$. It is not clear what is meant by "equivalent to ~30 % of the glaciers and ~89 % of the glacierised area" (line 14-15).**

Response: By that sentence we meant that out of the entire glaciers of studied region, glaciers >0.5 km$^2$ comprise ~30% of the total glacier population and ~89% of the glacierised area. These figures are based on the total number of glaciers reported by RGI 6.0 and our study. We have now revised the sentence accordingly.

*"The study records data on 2257 glaciers (of individual size >0.5 km2) covering an area of ~7923 ±106 km2 which is equivalent to ~30% of the total glacier population and ~89% of the total glacierised area of the region."*

**3: The term "deglaciation" is used two times in line 19 and could be replaced by area loss.**

Response: We have removed the sentence as per the Editors comments.

**4: In the last part of the abstract (line 20-23) the authors should inform about the type of climate data used and the length of the observation period.**

Response: We have removed the section on the climate data analysis as per the Editors comments.

**Introduction**

**5: Many articles on Himalayan glaciers begin with this kind of introduction like … third pole, water tower for large populations in adjoining lowlands, and so forth. However, the specific characteristic of glaciers in the cold-arid region of Ladakh should be highlighted in this context.**

Response: We have now revised the section accordingly.

*"Any change to the Himalayan cryosphere would have a direct impact on the hydrology, further influencing the communities downstream whose livelihood and economy relies on, and are supported by, the major river systems e.g., the Brahmaputra, Ganges and Indus, among others. In high altitude arid regions like Ladakh, where the majority of glaciers are small and restricted to higher altitudes, meltwater serves as an important driver of the economy, especially in years with low winter precipitation when glacier melt becomes the major (or only) source of water (Schmidt & Nüsser, 2012, 2017)."*

**6: Some of the urban agglomerations (Dhaka, Kolkata, Karachi) are less or not at all dependent on the glaciers of Ladakh. It might be more important to refer to basic studies on socio-hydrological interactions in Ladakh and the direct problems of water scarcity for irrigated cultivation (e.g. Nüsser et al. 2012 in Mountain Research and Development).**

Response: We have removed the sentence and included more information specific to Ladakh region. See the response to the comment 5 above.

**7: Line 27: What is the meaning of hydro-economy. This term is a bit vague and can mean different things including hydropower generation and irrigated crop cultivation.**

Response: Hydro-economy means the economy that revolves around water which includes agricultural irrigation, domestic consumption, industrial use and hydro-power generation among others. Himalaya is large and there are several sectors that make use of the water coming from the region, hence the term "hydro-economy".

**8: Line 28 it should read "Himalayan cryosphere"**

Response**:** Corrected.

**9: Line 40 it should read ", not all regions of Ladakh". In this context the authors should mention glacier studies that have been conducted in Ladakh. Maybe at the end of this paragraph (line 43).**

Response: We have revised the section accordingly.

*"However, not all regions of Ladakh have been analysed at the same level of spatio-temporal detail. In particular, our knowledge of glacier dynamics and their response to climate change is still incomplete in the cold-arid, high-altitude Ladakh region (~105,476 km2) comprising both, the Himalayan and Karakoram ranges. Few studies have focused on the glaciers of this region (e.g. Bhambri et al., 2011, 2013; Chudley et al., 2017; Negi et al., 2021; Nüsser et al., 2012; Schmidt & Nüsser, 2012, 2017; Shukla et al., 2020). "*

**10: Line 47-48 The inventories by GSI and SAC are manually demarcated. "among others" needs references.**

Response: We have checked the sources again and found that GSI inventory (GSI 2009) and SAC inventory (SAC 2011) was developed with the help of Survey of India toposheets of 1:50000 scale with contour interval of 40m or degree sheets maps on 1:250000, whichever were available with combined aid of aerial photography and satellite imagery. However, the methodology used is not clear, whether it was developed through automated, semi-automated or manual approach, for these inventories. Therefore to avoid any confusion, we have now revised the manuscript accordingly and also provided references for the term "among others".

*"Glacierised area estimations have often relied on global and regional glacier inventories such as the Randolph Glacier Inventory (RGI), Global Land Ice Measurements from Space (GLIMS), Geological Survey of India (GSI) inventory and Space Application Centre India (SAC) inventory, among others (Chinese Glacier Inventory (CGI), Glacier Area Mapping for Discharge from the Asian Mountains (GAMDAM), International Centre for Integrated Mountain Development (ICIMOD)). However, given the large scale of these inventories, automated techniques are employed, in most of the cases, to map and calculate glacier extent with differing levels of success. Additionally, the varying quality of satellite imagery acquired from different time periods are sometimes necessitated in high mountain areas, such as Ladakh."*

**11: Line 50: why necessitated?**

Response: Additional imageries of varying qualities are required because of majorly two reasons i.e. cloud cover and seasonal snow which is frequent in high altitudes of Himalaya.

**12: Line 52: it should read "entire Ladakh region"**

Response: Corrected.

**13: Line 62: it should read images or imagery**

Response: Corrected

**14: Line 65-67. This sentence is important and should come earlier in the introduction.**

Response: Rearranged

**15: Line 67: instead of arid season, lean or dry season might be more appropriate.**

Response: Corrected

**16: Line 68: it should read "can be viewed and downloaded from…"**

Response: Corrected

Study area

**17: The main problem is the unclear use of the terms UIB, Ladakh and study area. This should be consistent from the title and abstract to the conclusions. The study focuses on the upper part of the UIB above Skardu. The title focuses on the Ladakh region. In some section of the article spatial denominations are not consistent. Do the authors refer to entire UIB in line 83? In a later section (line 132) the authors state "…the entire UIB, upstream of Skardu, was investigated."**

Response: UIB in the manuscript refers to the Upper Indus Basin area upstream of Skardu region only, which we have now cleared at the beginning of the section 2: Study area.

*"This study focuses on glaciers in the Upper Indus Basin (UIB) upstream of Skardu and three internal drainage/endorheic basins (IDBs) within Ladakh, namely Tsokar, Tsomoriri and Pangong Basins."*

We have now revised the terms accordingly to avoid further confusion. The combined area of UIB and IDBs are now referred to as Ladakh region throughout the manuscript as majority of the area falls under the Ladakh region. However, individually they are still referred to as UIB or IDBs as they are the subsets within the region.

*"Since the majority of the investigated area (UIB and IDBs combined) falls within Ladakh, the combined area of UIB and IDBs will be referred to as "Ladakh region" hereafter."*

We have also revised the line 132.

*"A small portion of the leftover area from UIB after second-order tributary basin delineation was merged into the Leh Basin in order to investigate the UIB upstream of Skardu. Delineation of the three endorheic basins (IDBs) that lie partially or completely in the Ladakh region, i.e., Tsokar, Tsomoriri and Pangong Basins, was also carried out using the same method with the help of respective lakes as a pour point. The digitisation of the three lakes (Tsokar, Tsomoriri and Pangong Lake) was carried out manually for the years 1977, 1994, 2009 and 2019 using Landsat imagery."*

**18: Figure 1: It might be informative to point out the location of the three endorheic basins (Tsomoriri, Tsokar and Pangong Tso) for those who are not familiar with the region. Pangong and Tsokar can be detailed in the same way as Tsomoriri. In Figure 2a this information is presented. In the figure caption it should read stars (line 77) because it is plural.**

Response: We have now revised the figure accordingly.

[Figure]

*Figure 1: Location map of the study area: the boundaries of studied Upper Indus Basin and internal drainage basins are outlined in black and red on the digital elevation model (DEM) and in the inset map. Inset map shows the study area with respect to the Himalayan and Karakoram region. Black dots and stars represent the respective basins' major settlements and field investigated glaciers. ASTER Global DEM was used to produce the base map.*

**19: Line 94-95: Census of India 2011 and Census of China cannot be found in the reference list**

Response: We have now added the sources to the reference list.

[revised manuscript text omitted]

**24: Line 183: Kanda et al. (2020) with brackets**

Response: The section has been removed following the editors comment.

**25: Line 194 to 203: Some grids have no ground stations for corrections. What is the range of bias?**

Response: The section has been removed following the editors comment.

**Results**

**26: The section should not begin with a table and a figure**

Response: Corrected.

**27: Figure 3: Although one has to expect the majority of glaciers on northern aspects, the complete absence of southern aspects needs to be checked. What is with the large valley glaciers like the Siachen glacier and some glaciers on the southern faces of Nun and Kun in western Ladakh?**

Response: We have rechecked our dataset concerning the aspect of glaciers in all the basins. We found our data in the figure to be accurate as per our analysis. In addition, the southern facing glaciers are not absent but relatively lower than those with other aspects. For example in case of Shayok basin, 8, 6 and 9% of the glacierised area has an aspect of South east, South and South west which is equivalent of 437, 350 and 471 $km^2$, respectively. In Zanskar and Pangong basins, the aspects are 6, 2 and 0.2%, and 19, 8 and 0.4% towards South east, south and South west, respectively. We have revised the manuscript accordingly

*"Around 74% (1665) of the glaciers face the northern quadrant (NW-NE) amounting to ~50% (3940 km2) of the glacierised area. While 9, 5, 3, 3 and 4% of the glaciers face East, South-East, South, South-West and West which constitute 24, 6, 8, 6 and 6% of the glacierised area, respectively. However, the orientation and respective area coverage of glaciers vary within individual basins (Figure 3i, ii)."*

**28: Figure 4: in the figure the term "glaciated area" is used and the figure caption uses "glacierized area". This must be consistent.**

Response: The section has been removed following the editors comment.

**29: In several parts (line 237, line 248, line 258) the authors use the term "Deglaciation". Area loss might be more appropriate. (later again in lines 350 and 360)**

Response: Correction is done wherever necessary.

**30: Lines 271 and 276: It should read "…between XXX and XXX%" )**

Response: The section has been removed following the editors comment.

**Discussion**

**31: The heading "Description …" is unusual in the discussion.**

Response: The subheading has now been revised

**32: Line 316: It should read "open-source file format"**

Response: Corrected.

**33: Line 321: "Jawaharlal Nehru University and University of Aberdeen glacier IDs" is not clear.**

Response: the "Jawaharlal Nehru University and University of Aberdeen glacier IDs" are the new glacier Ids that were assigned to the glaciers of this new inventory. We have now revised the sentence for more clarity.

*"For each region, there is one file for basin outlines, and four files for glacier and lake (if present) outlines for 1977, 1994, 2009 and 2019. Each glacier outline file contains glacier Ids (New glacier Ids, Randolph Glacier Inventory 6.0 Ids, and Global Land Ice Measurements from Space initiative Ids), coordinates (latitude and longitude), elevation (maximum, mean and minimum), aspect (mean), slope (mean), area, length (maximum), area uncertainty and length uncertainty. Whereas, the Lake Outline file contains coordinates, area, elevation and area uncertainty. "*

**34: In the comparison with recent studies (section 5.4), the authors should also refer to the study by Bhambri et al. 2013: Heterogeneity in glacier response in the upper Shyok valley, northeast Karakoram**

Response: We have revised the section accordingly and compared our results with Bhambri et al 2013 as well.

**35: Some brackets are needed in lines 404, 408, 422**

Response: Corrected.

Conclusions

**36: Line 448: What are "more favorable climatic conditions"? For agriculture or for plant growth?**

Response: The section has been removed following the editors comment.

References

**37: L 510: Reference Frey et al. 2014 title needs to be corrected**

Response: Corrected

---

## Author Comment (AC2)

**RESPONSE TO REFEREE#2**

We thank the reviewer#2 for the review. We have carefully modified the manuscript according to the comments and suggestions. Below, we provide our responses (normal texts) to the comments (**Bold texts**) made by the reviewer#2. The *italics texts* are used to highlight the specific changes that were made in the manuscript.

The revised dataset can be viewed and downloaded using the following link: https://www.pangaea.de/tok/6d533482a662ef2124ed91eabdeec7b358dd8058

**1: Data Section 3.1, line 101: Landsat datasets used for multitemporal inventory for the year 1977 varies for +-5 years. For the other years variations is 1 year, which is understandable considering cloud and snow conditions. However, knowing the status of the change of snout of glaciers is it justifiable to consider one result from datasets with 5 years span period.**

Response: Thank you for the comment.

We have used a total number of 17 scenes from 1972-1980, out of those 12 are from the year 1976-1977, 3 from 1979-1980 and 2 from 1972. However, 3/5 scenes outside the year 1976-1977 has been used to aid the digitization process as the same scene also exists for the year 1976-1977 (TABLE S1). The remaining scene, from outside 1976-1977 period, was used to digitize 14/246 (5%) glaciers of Leh basin and 37/256 (15%) glaciers of Zanskar basin. Overall, the scenes from outside 1976-1977 period was only used for a fraction of glaciers of Ladakh region i.e. 51/2257 (2%) glaciers.

We believe that this will have an impact on the results to some extent in individual basin even though the numbers are quite small. However, availability of the data from earlier Landsat period in this region is rare. Therefore, the dataset from 1977±5 should be acceptable and well justified.

We have revised the section for more clarity

*"This study utilises multiple Landsat level-1 precision and terrain (L1TP) corrected scenes (63 scenes in total) from four different periods: 1977±5 (hereafter 1977), 1994±1 (hereafter 1994), 2009 and 2019±1 (hereafter 2019). Scenes from the 1970s are majorly (12 out of 17) from the year 1976 and 1977 however due to higher cloud cover and less availability of imagery during the earlier Landsat period, five scenes from 1972, 1979 and 1980 were also included to aid the digitization of glaciers (Table S1)."*

**2: Glacier Mapping, section 3.3., line 162-163: authors have fixed map scale for digitization to reduce error. Landsat is a medium resolution data, which can produce a map of scale of 1:25,000. Pan sharpen data will give larger scale map. Zooming to 1:10,000 may distort pixels which can make the digitization a little difficult. Setting 1:10,000 map scale for MSS data having 60m spatial resolution will further have deterioration. How authors have dealt with this minor but practical issues related to digitization should be made clear.**

Response: We have used a fixed scale of 1:10000 for the majority of the scenes that involve larger glaciers and our analyst didn't face any difficulty in digitization. However for MSS and smaller glaciers, we have also used 1:25000 whenever required.

We have revised the section for more clarity on this.

*"Furthermore, some mapping errors are still expected to be present in this inventory due to a possible misinterpretation of glacier features, and the quantification of such errors are difficult owing to the lack of reliable reference in-situ data in the Ladakh region. Such errors were minimized by keeping a fixed map-scale of 1:10,000 in most cases, and undertaking a quality check on glacier outlines using high-resolution images. In case of MSS images and smaller glaciers, a map-scale of 1:25,000 was also used whenever required."*

---

## Author Comment (AC3)

**RESPONSE TO EC1**

We thank the editor for the time in reviewing the manuscript. We have carefully modified the manuscript according to the comments and suggestions. Below, we provide our responses (normal texts) to the comments (**Bold texts**) made by the editor. The *italics texts* are used to highlight the specific changes that were made in the manuscript.

The revised dataset can be viewed and downloaded using the following link: https://www.pangaea.de/tok/6d533482a662ef2124ed91eabdeec7b358dd8058

1. **ESSD does not publish science papers, and there is a significant amount of science in your manuscript. It begins on L18 when you discuss deglaciation trends and then L20 when you discuss external factors contributing to the deglaciation.**

Response: We have made the necessary changes and the manuscript is now revised accordingly.

2. **The ERA5 modeling effort should be removed. Sections 3.5, 4.5, 5.2 and the middle two paragraphs of 7.Conclusion should be removed.**

Response: We have removed these sections in the revised manuscript.

3. **Even sections 4.2, 4.3, 4.4 could be removed as this trend analysis is external to the data product, where more focus is needed. However, if these sections stay I am not opposed to it.**

Response: We have also removed the required sections as per your suggestions. We have also made new additions to the results focusing on data only. The new additions include (4.1) General statistics, (4.2) Glacier distribution in the Ladakh region and (4.3) Glacier hypsometry, slope and aspect.

*"4.1. General statistics*

*"In total, 2257 glaciers (>0.5 km2) were compiled in the current inventory (Table 2), with a total glacierised area of ~8511±430, 8173±215, 8096±214 and 7923 ±106 km2 for the years 1977, 1994, 2009 and 2019, respectively. The glacierised area corresponds to ~6% of the Ladakh region with individual areas ranging between 0.5±0.02 and 862±16 km2. Glacier length in the Ladakh region varies between 0.4±0.02 and 73±0.54 km with a mean length of 2.9±0.05 km. About 90% of the glaciers are shorter than 5km in length while only 6% of glaciers have a length of < 1km. Larger glaciers are mainly located in the Shayok and Zanskar Basins with the Siachen Glacier being the largest (862±16 km2), longest (73±0.54 km) and covers the greatest elevation range of ~3616m (3702-7318m a.s.l.). The major lakes in each endorheic basins of Pangong, Tsokar and Tsomoriri occupy an area of 3, 2 and 2.5%, respectively. The lake areas for the year 1977, 1994, 2009 and 2019 were 610±14, 619±8, 669±8 and 705±8 km2 for Pangong, 13.5±0.9, 17±0.7, 18.3±0.7 and 18.8±0.6 km2 for Tsokar and 140±2.6, 141±1.3, 141±1.3 and 141±1.1 km2, respectively."*

4.2. Glacier distribution in the Ladakh region

*"Glacierised areas and population in the Ladakh region vary across basins. Shayok Basin has the largest distribution of glacierised area and population (74% and 56%), whereas the Tsokar Basin has the least (0.04% and 0.1%), respectively (Table 2). Based on size distribution, the glacier area category of 1-5km2 comprises the highest area (28% of the total), while the category of 50-100km2 occupies the least glacierised area (9%) of the region. Most glaciers (~90% of the total) in the Ladakh region have an area of <5km2 but occupy only 37% of the total glacierised area. The population and area of glaciers in each area class are different in each basin but the proportion of glaciers, smaller than 5km2, is greater than 87% in all basins. Glaciers larger than 100 km2 (n=7, < 1% of the total) are only present in the Shayok Basin and occupy ~24 and 32% of the total glacierised area of Ladakh and Shayok Basin, respectively.*

**4.3.** Glacier hypsometry, slope and aspect

*"Figure 3 (iii and iv) shows the glacier elevations and hypsometry with 100m elevation intervals of seven basins of the Ladakh region. The highest and lowest glacier elevation are 7740 and 3249m a.s.l., both in the Shayok Basin. Whereas mean elevation of the glacier ranges between 4345-6355m a.s.l. (Figure 3iii). Small glaciers mainly occupy the higher elevations above 5500, and vice versa. The majority (73%, 5810 km2) of the glacierised area is distributed in the 5000-6000*

*m a.s.l. elevation range, while only 14% is located below 5000m, and 13% above 6000m a.s.l. (Figure 3iv). The mean slope of these glaciers ranges between 8 and 46º, and is found to decrease with increasing glacier area. Glaciers with an area greater than 100 km2 (n- 7, <1% of the total) have the lowest mean slope of 13º whereas, higher mean slopes (23º) are found for smaller glaciers (43% of the total). Overall, the mean glacier slope is ~21º (Figure 3v). Around 74% (1665) of the glaciers face the northern quadrant (NW-NE) amounting to ~50% (3940 km2) of the glacierised area. While 9, 5, 3, 3 and 4% of the glaciers face East, South-East, South, South-West and West which constitute 24, 6, 8, 6 and 6% of the glacierised area, respectively. However, the orientation and respective area coverage of glaciers vary within individual basins (Figure 3i, ii)."*

5. **In Section 3.4 you attribute uncertainty to satellite resolution, but it seems like there may be other factors that contribute to uncertainty than only this.**

Response: We understand that there might be several other factors that influence the outlines, such as a shift between temporal satellite scenes (orthorectification). Although, most of these are difficult to quantify systematically, also due to the lack of reliable reference data (Racoviteanu et al., 2009; Shukla et al., 2020), they tend to be rather small. For example, the uncertainties arising from orthorectification are negligible (Heid and Kääb, 2012). Nonetheless, we have now revised the uncertainty section and have improved uncertainty estimation and reporting, where possible. The uncertainty for glacier area, glacier length and lake area, through buffer-based approach, are now included in the dataset. The widely used buffer-based approach gives more realistic results when it comes to a large set of glaciers as it varies based on the size of the glacier and the quality of the images (Mölg et al., 2018; Paul et al., 2017).

*"This study involves the use of satellite imagery to extract various glacier parameters. It is therefore subject to uncertainties which may arise mainly from four different sources: (1) the quality of the image (with potential issues due to seasonal snow, shadows and cloud cover), (2) sensor characteristics (spatial/spectral resolution), (3) interpretation of glacial features and methodology used, and (4) post-processing techniques (Le Bris & Paul, 2013; Paul et al., 2013, 2017; Racoviteanu et al., 2009, 2019). Error due to sources 1, 3, and 4 are generally minor and can be visually identified and corrected (section 3.3), but an exact quantification is difficult due to*

*the lack of reference data available from the region (Racoviteanu et al., 2009; Shukla et al., 2020). Type 4 errors are significant and have an impact on both glacier area and length estimation. Therefore, we applied a buffer-based assessment to glacier areas with the buffer width set to one-pixel for debris covered and a half-pixel for clean ice (Bolch et al., 2010; Granshaw & Fountain, 2006; Mölg et al., 2018; Paul et al., 2017; Racoviteanu et al., 2009; Shukla et al., 2020; Tielidze & Wheate, 2018), given that the level 1TP Landsat images were corrected to sub-pixel geometric accuracy (Bhambri et al., 2013). A buffer-based method provides the maximum and minimum estimates of uncertainty with respect to glacier size, where the values vary with size of the glacier and spatial resolution of the imagery used. Thus, it is more specific to the dataset and most recommended when there is no reliable reference data available (Paul et al., 2017; Racoviteanu et al., 2009; Shukla et al., 2020). The same approach was also followed to estimate the uncertainties in lake areas with one-pixel as the buffer width.*

*The associated uncertainty for smaller glaciers (<0.5 km2) amounts to ~12-25%. Therefore, all the glaciers with an area of less than 0.5 km2, which comprise ~70% and ~10% of the total glacier count and glacierised area respectively, are not included in this study. For the remaining glaciers, the uncertainty in glacier area ranged between ±2.1 and ±7.2% depending on the spatial resolution of the satellite imagery and the individual glacier size. The highest uncertainty was for the year 1977 due to the coarser spatial resolution of Landsat MSS data when applied to the smallest glaciers (0.5-1 km2). For most of the glaciers, lengths are assumed to be accurate to ±1 pixel at the terminus (Le Bris & Paul, 2013). Therefore, a buffer of one-pixel was set to determine the uncertainty in glacier length. The length uncertainty ranged between ±1.5 and ±2.6% with maximum uncertainty observed for the smallest glacier category (0.5-1 km2). The methods yielded an overall uncertainty of 4.2, 1.8 and 1.5% for glacier area, glacier length and lake area, respectively (Table S2).*

*Uncertainties related to other attributes (mean elevation, mean slope and mean aspect) of the inventory are difficult to estimate due to the use of the ASTER GDEM product in this study, which was developed using a collage of archived scenes acquired between 2000 and 2013. In addition, the local undulations and surface change over time will have only marginal effects on parameters (elevation, slope and aspect) that are averaged over the entire glacier as averaging compensates for most of the changes (Frey & Paul, 2012). However, for parameters like maximum and*

*minimum elevations, where one cell is used and no averaging is applied, the uncertainty is ~ ±9m, as the vertical accuracy of ASTER GDEM is ±8.55m for glacierised areas of high Asia (Yao et al., 2020) and ±8.86m elsewhere (Mukherjee et al., 2013)."*

**6. In Section 5.3 you compare to other inventories and have justification for various differences - but are all of the errors attributed to the other inventories, and yours is only uncertainty based on satellite resolution?**

Response: We have now revised the section in the manuscript accordingly. Our approach was similar to GAMDAM and ICIMOD inventories, where they have used a Normalized Standard Deviation approach on the outlines/datasets produced by several operators on same subset of glaciers. Their approach is also influenced majorly by the satellite resolution. According to Paul et al., 2017, the method used by GAMDAM and ICIMOD gives more realistic estimate than the buffer-based approach if it is performed on the entire population of glaciers in a dataset, rather than on a subset of glaciers. However, the approach requires higher workload (digitization of all the glaciers by several analyst) and it would be too time consuming to apply on the current inventory which deals with >2000 glaciers in four time periods each. For RGI inventory, an entirely different approach on error estimation was adopted than the present, GAMDAM and ICIMOD approaches due to the fact that the data produced in RGI is global, deals with an extremely large set of glaciers which were sourced from different analysts involving different methodologies. Therefore, we believe that the uncertainty based on the buffer-based approach is one of the best to use in the present study. We have now included the following text in the manuscript to clarify this:

*"The comparison involves glacier outlines for 2009 from the present study and excludes glaciers smaller than 0.5 km2 from regional inventories to achieve the closest match temporally and for glacier size categories. This should be taken as a first order comparison, given the fact that the uncertainties have been estimated with different approaches for the different inventories. Specifically the uncertainty estimated for the GAMDAM and ICIMOD inventories differs only slightly to the one applied here, given that they used a normalized standard deviation approach on the datasets produced by several operators on the same subset of glaciers (Bajracharya et al.,*

*2011, 2019; Guo et al., 2015; Nuimura et al., 2015; Sakai, 2019). Whereas, in case of RGI 6.0 inventory, the uncertainty estimation approach differs significantly from the one presented here, because their errors were calculated on a collection of glaciers due to the vast quantity of data acquired from multiple sources and approaches used to produced them (Pfeffer et al., 2014)."*

**7. For section 5.4 can you take the other recent studies you compare against, and compare like with like? That is, limit them (or you) to the same subset of glacier size, and then quantitatively compare both some individual glaciers and the bulk properties of the two datasets? This could then lead to a significantly more robust validation discussion and uncertainty results.**

Response: We have now revised the manuscript according to the comments. We have also tried to compare like with like. However, comparing with the exact location or data or time period was difficult as all the studies undertaken in the Ladakh region have either different time periods, or do not have the data in the public domain (except the stats from the literature and Shukla et al., 2020's entire dataset: https://doi.pangaea.de/10.1594/PANGAEA.904131). We have also undertaken a comparative analysis on 21 individual glaciers from across the region and the bulk properties of the dataset. This additional analysis was quite helpful and we have now included two figures and a supplementary table, detailing how our glacier outlines compare with other studies. We have also revised the section:

*"The data from the recent spatio-temporal change studies from different sub-regions of Ladakh (Figure 5) are not in the public domain, except from Shukla et al., 2020. Hence, it is not possible to use these to validate our results. Therefore, our comparison mostly focuses on the rate of change for some of the individual glaciers (n=21, Figure 5) from the literature and the bulk properties of a set of glaciers in different regions (Table S3, Figure 6). Our results agree well with the studies conducted by others (Bhambri et al., 2013; Chudley et al., 2017; Garg et al., 2022; Garg et al., 2021; Negi et al., 2021; Schmidt & Nüsser, 2012, 2017; Shukla et al., 2020) on individual glaciers of various sizes as well as on a set of glaciers, respectively (Figure 6, Table S3). However, the results differ significantly only on some glaciers and especially in a part of the Shayok Basin (e.g.*

*Kumdan (D), Aktash (E) and Thusa glaciers(I)). In the Shayok Basin surge-type glaciers are common (Bhambri et al., 2013, 2017), the difference in analysis period between the present and other studies is the likely cause of the difference in glacier area statistics. Figure S2 presents the dynamics of the Kumdan and Aktash glaciers as an example of surge type glacier of this region.*

*No significant difference was observed in rate of change of glacierised areas between the present study and other studies in the Leh, Tsomoriri, Zaskar and Suru Basins. In contrast, the number of glaciers and glacierised area vary among these studies (present and others) but paint a similar picture of relatively lower retreat in the Shayok Basin (Bhambri et al., 2013; Negi et al., 2021), higher in Leh, Tsokar, Tsomoriri (Chudley et al., 2017; Schmidt & Nüsser, 2012, 2017) and moderate in Zanskar and Suru Basins (Garg et al., 2022; Garg et al., 2022; Shukla et al., 2020)."*

*"*

[Figure]

*Figure 5: Presents the spatial extent of different studies undertaken in Ladakh region. Black stars represent the individual glaciers.*

[Figure]

*Figure 6: Comparison between the present study and other studies undertaken in different basins of Ladakh region over different time periods*

*Table S3: Comparison of glacier change attributes between the present study and others recent studies.*

| Source | Basin | Range/ Glacier/ | Code | Other recent studies | | | | | Present study | | | | |
|---|---|---|---|---|---|---|---|---|---|---|---|---|---|
| | | | | No. of | Size (km²) | Year | Total | Retreat rate | No. of | Size (km²) | Year | Total | Retreat |
| Negi et al., 2021 | Shayok | Shayok sub-basin | SSB | 569 | >1 | 1991-2014 | -0.19 | -0.008 | 603 | >1 | 1994-2019 | -0.94 | -0.038 |
| | | Siachen Glacier | A | 1 | >100 | 1990-2014 | -0.08 | -0.003 | 1 | >100 | 1994-2019 | -0.28 | -0.007 |
| | | Central Rimo Glacier | B | 1 | >100 | 1990-2014 | -0.52 | -0.022 | 1 | >100 | 1994-2019 | 0.61 | 0.015 |
| | | Southern Rimo Glacier | C | 1 | >100 | 1990-2014 | -1.05 | -0.044 | 1 | >100 | 1994-2019 | -1.87 | -0.045 |
| | | Mamosto Glacier | F | 1 | 50-100 | 1990-2014 | -0.17 | -0.007 | 1 | 50-100 | 1994-2019 | -0.79 | -0.019 |
| | | Kumdan Glacier | D | 1 | 50-100 | 1990-2014 | -2.35 | -0.098 | 1 | 50-100 | 1994-2019 | -0.84 | -0.020 |
| | | Urdolep Glacier | J | 1 | 10-50 | 1990-2014 | -0.19 | -0.008 | 1 | 10-50 | 1994-2019 | -0.59 | -0.014 |
| | | Layong ma Glacier | H | 1 | 10-50 | 1990-2014 | -1.71 | -0.071 | 1 | 10-50 | 1994-2019 | -2.39 | -0.057 |

| Reference | Region | Glacier | Code | N | Size | Period | V1 | V2 | N | Size | Period | V3 | V4 |
|---|---|---|---|---|---|---|---|---|---|---|---|---|---|
| | | Lagongma Glacier | G | 1 | 10-50 | 1990-2014 | -0.277 | -0.011 | 1 | 10-50 | 1994-2019 | -1.47 | -0.035 |
| | | Aktash Glacier | E | 1 | 10-50 | 1990-2014 | 16.88 | 0.703 | 1 | 10-50 | 1994-2019 | 2.39 | 0.057 |
| | | Thusa Glacier | I | 1 | 10-50 | 1990-2014 | -7.47 | -0.311 | 1 | 10-50 | 1994-2019 | -1.81 | -0.043 |
| Bhambri et al. 2013 | | Upper Shayok Basin | USB | 136 | >0.2 | 1974-2011 | 0.14 | 0.004 | 570 | >0.5 | 1977-2009 | -2.2 | -0.069 |
| | | Central Rimo Glacier | B | 1 | >100 | 1974-1998 | 0.04 | 0.002 | 1 | >100 | 1977-1994 | -1.88 | -0.111 |
| | | Central Rimo Glacier | B | 1 | >100 | 1998-2011 | -0.59 | -0.045 | 1 | >100 | 1994-2009 | 1.16 | 0.078 |
| | | Kumdan Glacier | D | 1 | 50-100 | 1974-1998 | -6.71 | -0.279 | 1 | 50-100 | 1977-1994 | -3.24 | -0.191 |
| | | Kumdan Glacier | D | 1 | 50-100 | 1998-2011 | 5.27 | 0.406 | 1 | 50-100 | 1994-2009 | 2.98 | 0.199 |
| | | Aktash Glacier | E | 1 | 10-50 | 1974-1998 | 0.37 | 0.016 | 1 | 10-50 | 1977-1994 | -0.78 | -0.048 |
| | | Aktash Glacier | E | 1 | 10-50 | 1998-2011 | 2.99 | 0.230 | 1 | 10-50 | 1994-2009 | 3.42 | 0.228 |
| Schmidt and Nüsser, 2017; | Leh/ Zanskar/ | Selected regions of central | CEL | 1800 | >0.03 | 1969-2016 | -19 | -0.4 | 517 | >0.5 | 1977-2019 | -19.3 | -0.46 |

| | | and eastern Ladakh | | | | | | | | | | | |
|---|---|---|---|---|---|---|---|---|---|---|---|---|---|
| | | Phuche Glacier | K | 1 | 0.5-1 | 1969-2016 | -18.0 | -0.50 | 1 | 0.5-1 | 1977-2019 | -21.0 | -0.50 |
| | | Hemis Shukpac hen Glaciers | L | 5 | 0.2-1 | 1969-2016 | -38.0 | -0.80 | 2 | 0.5-1 | 1977-2019 | -33.0 | -0.78 |
| | | Stok Range | M | 7 | 0.2-1 | 1969-2016 | -22.4 | -0.40 | 3 | 0.5-1 | 1977-2019 | -10.5 | -0.30 |
| | | Kang Yatze | N | 35 | 0.5-1 | 1969-2010 | -18.8 | -0.40 | 26 | 0.5-1 | 1977-2010 | -24.0 | -0.50 |
| | | Kang Yatze | N | 25 | >1 | 1969-2010 | -12.2 | -0.20 | 17 | >1 | 1977-2010 | -11.0 | -0.25 |
| | | Lungser Range | O | 39 | 0.5-5 | 1969-2014 | -17.7 | -0.40 | 22 | 0.5-5 | 1977-2019 | -16.4 | -0.39 |
| Chu dley et al, 201 7 | Leh/ Shayok | Central Ladakh range | CL R | 76 | 1-5 | 1991-2014 | -6.6 | -0.29 | 82 | 1-5 | 1994-2019 | 7.1 | -0.39 |
| Shukla et al., 2020 | Suru | Suru Sub-basin | SU B1 | 13 0 | >0.5 | 1971-2017 | -9.08 | -0.20 | 13 6 | >0.5 | 1977-2019 | -14 | -0.28 |

| | | | | | | | | | | | |
|---|---|---|---|---|---|---|---|---|---|---|---|
| Suru Sub-basin | SUB2 | 22 | 0.5-1 | 1971-2017 | -24.04 | -0.52 | 22 | 0.5-1 | 1977-2019 | -25.18 | -0.60 |
| Suru Sub-basin | SUB3 | 47 | 1-5 | 1971-2017 | -12.10 | -0.26 | 47 | 1-5 | 1977-2019 | -16.03 | -0.38 |
| Suru Sub-basin | SUB4 | 15 | 5-10 | 1971-2017 | -4.15 | -0.09 | 15 | 5-10 | 1977-2019 | -7.91 | -0.19 |
| Suru Sub-basin | SUB5 | 6 | 10-50 | 1971-2017 | -11.11 | -0.24 | 6 | 10-50 | 1977-2010 | -4.33 | -0.10 |
| Suru Sub-basin | SUB6 | 1 | 50-100 | 1971-2017 | -0.28 | -0.01 | 1 | 50-100 | 1977-2010 | -1.24 | -0.03 |
| Tongul Glacier | P | 1 | 5-10 | 1971-2017 | -11.07 | -0.24 | 1 | 5-10 | 1977-2019 | -6.14 | -0.15 |
| Rantak Glacier | Q | 1 | 5-10 | 1971-2017 | -15.23 | -0.33 | 1 | 5-10 | 1977-2019 | -11.52 | -0.27 |
| Sentik Glacier | R | 1 | 1-5 | 1971-2017 | -14.06 | -0.31 | 1 | 1-5 | 1977-2019 | -10.48 | -0.25 |
| Parkachik Glacier | S | 1 | 10-50 | 1971-2017 | -0.27 | -0.01 | 1 | 10-50 | 1977-2019 | -1.24 | -0.03 |
| Shafat Glacier | T | 1 | 10-50 | 1971-2017 | -15.57 | -0.34 | 1 | 10-50 | 1977-2010 | -14.18 | -0.34 |

| | | | | | | | | | | | | | |
|---|---|---|---|---|---|---|---|---|---|---|---|---|---|
| | | Dulung Glacier | U | 1 | 10-50 | 1971-2017 | -10.81 | -0.24 | 1 | 10-50 | 1977-2010 | -9.05 | -0.22 |
| | | Chilung Glacier | V | 1 | 5-10 | 1971-2017 | -12.74 | -0.28 | 1 | 5-10 | 1977-2019 | -15.20 | -0.36 |
| | | Lalung Glacier | W | 1 | 10-50 | 1971-2017 | -6.57 | -0.14 | 1 | 10-50 | 1977-2019 | -3.35 | -0.08 |
| | | Pensilungpa Glacier | X | 1 | 10-50 | 1971-2017 | -6.14 | -0.13 | 1 | 10-50 | 1977-2019 | -3.39 | -0.08 |
| Garg et al., 2022 | | Parkachik glacier | S | 1 | 10-50 | 1971-2018 | -3.30 | -0.07 | 1 | 10-50 | 1977-2019 | -1.24 | -0.03 |
| Garg et al., 2021 | | Pensilungpa Glacier | X | 1 | 10-50 | 1993-2016 | -2.59 | -0.11 | 1 | 10-50 | 1994-2018 | -0.94 | -0.04 |

**8. Glacier product does not have uncertainty for area, Lmax, slope, etc.**

Response: We have now added the uncertainty of area and Lmax to the individual glaciers across the basins and time periods of the dataset. The dataset is now revised with new additions i.e. U_Area (Uncertainty in glacier area) and U_Lmax (Uncertainty in maximum glacier length). Please see the response to the comment number 4 for more detail on this. The revised data can be accessed using the following link

https://www.pangaea.de/tok/6d533482a662ef2124ed91eabdeec7b358dd8058

9. **Lake data product does not have uncertainty. The lake product is not widely discussed in your manuscript that focuses on glaciers. Why is this included and why only at one time?**

Response: We have provided lake outlines as the study includes three endorheic basins. We have now extended the lake outlines for the four periods (1977, 1994, 2009 and 2019) including the attributes like coordinates, mean elevation, area and area uncertainties. We have also revised the manuscript accordingly and discussed the lakes in methods, uncertainty and results sections.

For example:

*"Delineation of the three endorheic basins (IDBs) that lie partially or completely in the Ladakh region, i.e., Tsokar, Tsomoriri and Pangong Basins, was also carried out using the same method with the help of respective lakes as a pour point. The digitisation of the three lakes (Tsokar, Tsomoriri and Pangong Lake) was carried out manually for the years 1977, 1994, 2009 and 2019 using Landsat imagery."*

Also see the response to comment number 4 for more detail on this.

---

## Author Response (AR2)

**Response to the Editor:**

**Comment:** Are supplemental tables included somewhere as CSV?

**Response:** Yes. Supplementary tables (in .csv format) are now added to supplementary files of the revised manuscript.

**Response to the "Notification to the authors":**

**Comment:** Please note that your reference list has not been compiled according to our standards. Please consider adjusting your reference list with the next revision of your manuscript.

**Response:** The references in the revised manuscript are now according to the journal standards.

**Comment:** Coloured or marked text in *.pdf supplement file is not allowed. Please provide a clean version of the *.pdf supplement files (with black text) for the next revision.

**Response:** The revised version of the supplementary file is now clean. We have also added the tables (Table S1, Table S2 and Table S3) in CSV format to the supplementary files.

**Comment:** Your "Abstract" does not contain a citation to the DOI.

**Response:** The DOI in the abstract and elsewhere now contains a citation (Soheb et al., 2022) in the revised manuscript.

**Comment:** I just noticed that your figures #4 and #5 contain a map. To clarify whether a copyright statement or a credit must be given in the map itself or in the caption.

**Response:** Our maps fall under case (b) and Imagery (Landsat, GDEM) we have used is distributed under the public domain. Therefore, we have added credit in the caption of the maps of the revised manuscript figures #1, #4 and #5.